# Human Gut Microbiota in Health and Selected Cancers

**DOI:** 10.3390/ijms222413440

**Published:** 2021-12-14

**Authors:** Aleksandra Sędzikowska, Leszek Szablewski

**Affiliations:** Chair and Department of General Biology and Parasitology, Medical University of Warsaw, ul. Chalubinskiego 5, 02-004 Warsaw, Poland; aleksandra.sedzikowska@wum.edu.pl

**Keywords:** gut microbiota, modulators of gut microbiota, dysbiosis, cancers, anticancer therapy

## Abstract

The majority of the epithelial surfaces of our body, and the digestive tract, respiratory and urogenital systems, are colonized by a vast number of bacteria, archaea, fungi, protozoans, and viruses. These *microbiota*, particularly those of the intestines, play an important, beneficial role in digestion, metabolism, and the synthesis of vitamins. Their metabolites stimulate cytokine production by the human host, which are used against potential pathogens. The composition of the microbiota is influenced by several internal and external factors, including diet, age, disease, and lifestyle. Such changes, called dysbiosis, may be involved in the development of various conditions, such as metabolic diseases, including metabolic syndrome, type 2 diabetes mellitus, Hashimoto’s thyroidis and Graves’ disease; they can also play a role in nervous system disturbances, such as multiple sclerosis, Alzheimer’s disease, Parkinson’s disease, and depression. An association has also been found between gut microbiota dysbiosis and cancer. Our health is closely associated with the state of our microbiota, and their homeostasis. The aim of this review is to describe the associations between human gut microbiota and cancer, and examine the potential role of gut microbiota in anticancer therapy.

## 1. Introduction

The gut microbiota comprises the organisms that live in the gastrointestinal (GI) tract. The adult human gut microbiota consists of 10^13^–10^14^ microorganisms/mL of luminal content, with their total weight estimated to be 1.5 kg [1]. The human gut microbiota is composed of bacteria, archaea, fungi, protozoa, and viruses, in which bacteria dominate. The genome of the human microbiota includes 100 times more genes than the human genome and 10 times more cells than the human body [2]. The human microbiota is characterized by three groups of bacteria, *viz.* symbionts, commensals and pathobionts, which coexist in a stable balance in healthy humans. Symbionts possess a health-promoting effect, commensals demonstrate a neutral effect, i.e., no positive and no negative, whereas pathobionts have the potential to induce pathology [3].

Human microbiota, especially those of the gut, may have a beneficial effect on human health, and may be involved in serious metabolic diseases, autoimmune diseases, and those of the human nervous system. Gut microbiota may also play an important role in the development and progression of cancer disease.

The role of gut microbiota in human health is investigated and described in different aspects. Gut bacteria synthesize the compounds necessary for human health, such as vitamins and amino acids, short-chain fatty acids, and secondary bile acids. These bacteria also synthesize neurotransmitters and neuropeptides, as well as playing a protective function. On the other hand, changes in the composition of gut microbiota, called dysbiosis, may cause several diseases. The correlation between dysbiosis and metabolic diseases and psychiatric diseases or disorders is well described in literature. Less is known on dependence on changes of gut microbiota in the development and/or progression of cancers. Moreover, the literature on the role of gut microbiota as a target in anti-cancer therapy is very poor. Therefore, we describe how gut microbiota may be associated with cancer diseases. We present more recent and important results obtained in research involved in this problem.

The dependence between microbiota and health, as well as disease, was observed a long, long time ago. Hippocrates, Father of Medicine, in fourth century BC, wrote “All disease begins in the gut”. Louis Pasteur, French chemist and microbiologist, honored as the Father of Bacteriology, and as the Father of Microbiology, has seen the important influence of bacteria on human health and disease. Therefore, he wrote “Gentlemen, it is the microbes who will have the last word” “Messieurs, c’est les microbes qui auront the dernier mot”. René Dubos, a French-American microbiologist, also observed the abovementioned dependence. He wrote “The states of health or disease are the expression of the success or failure experienced by the organism in its efforts to respond adaptively to environmental challenges”. René Dubos is often attributed as the author of the environmental maxim: “Think globally, act locally”.

The idea that microbes can promote the development of cancers is old, and acquired a certain popularity in the late 1980s and early 1900s in the context of experiments and observations that were later disproven. Several recent studies have analyzed the oral and gut microbial dysbiosis associated with cancer risk and progression. In this review, we highlight studies analyzing the role of gut microbiota in human health and diseases, especially in cancer cells. We describe the association between changes of human gut microbiota and the development of selected cancers. We also discuss various strategies for microbial modulation. Additionally, we also summarize the potential role as a target in anticancer therapy.

## 2. Composition and Distribution of Human Microbiota

The human gut contains approximately 1000 different species of bacteria. To date, 50 phyla of bacteria have been described [4], four of which dominate in human gut microbiota: Bacteroidetes (Gram-negative bacteria), Firmicutes (Gram-positive, aerobic and anaerobic bacteria), Proteobacteria (for example the genera *Escherichia* and *Enterobacter*), and Actinobacteria (for example *Bifidobacterium*). Of these, the predominant phyla, accounting for up to 70 to 90% of total bacteria, are Firmicutes (60–80%), with more than 200 genera, the most important being *Ruminococcus*, *Clostridium*, *Eubacterium*, *Faecalibacterium*, *Roseburia*, and *Mycoplasma*, and the Bacteroidetes (20–30%), with the genera *Bacteroides*, *Prevotella*, and *Xylanibacter* [5]. Of the remainder, the Actinobacteria and Proteobacteria both constitute less than 10, and the minor phyla Fusobacteria and Verrucomicrobia occur relatively rarely [6,7]. In addition to the human gut, they are known to colonize various compartments of the human body. For example, various lipophilic bacteria, such as *Propionibacterium* spp. and *Malassezia* spp., are present in human skin [8,9].

A number of fungi also colonize the human body. The human gastrointestinal tract has been found to contain 66 fungal genera, including 184 fungal species [10]. In addition, the oral cavity is known to contain fungi belonging to the phylum Ascomycota, such as *Candida* spp., *Fusarium* spp., as well as various species of *Saccharomycetaceae*, such as *Candida albicans*, *C. dubliniensis*, *C. rugose* and *C. pararugosa*, as well as *Saccharomyces cerevisiae*, *Hanseniaspora uvarum* and *Pichia* spp. [11,12]. The human gastrointestinal tract is home to several fungal species from the genera *Candida*, *Saccharomyces*, *Aspergillus*, *Cryptococcus*, *Malassezia*, *Cladosporium*, *Galactomyces* and *Trichosporon* [13]. Other studies have also reported the presence of the genera *Candida*, *Aspergillus*, *Penicillium*, *Eremothecium*, *Trichospora*, *Cladosporium*, *Epicoccum*, and *Cryptococcus* in the human mouth, *Candida* in the stomach, and the genera *Aspergillus*, *Malassezia*, *Trichosporon*, *Eremothecium*, *Cladosporium*, *Saccharomyces*, *Candida*, *Cryptococcus* and *Galactomyces* in the intestine, between the duodenum and colon [14,15]. Species of *Aspergillus*, *Penicillium*, and *Candida* have been identified in the lungs of healthy humans [11,12,16], and *Candida albicans*, *C. glabrata*, *C. krusei*, *C. parapsilosis* and *Saccharomyces cerevisiae* in the vagina of healthy women are low [17,18].

The composition of the human microbiota is not homogenous throughout the body; its precise makeup at any location depends on a range of factors such as the organ, region of the GI or type of bacteria, among others (Table 1). It is also strongly influenced systematically by the mode of delivery at birth, age, sex, ethnicity, diet, diseases, use of antibiotics and other pharmaceuticals, as well as probiotics and prebiotics. It is also important to consider that the bacteria of the human microbiota may play various roles, and they can be beneficial as well as pathological (Table 2).

## 3. The Role of Gut Microbiota in Human Health

The human gut microbiota is involved in several processes that influence human health. Its microbiological components synthesize a range of compounds necessary for human health, such as vitamins, amino acids, and several other small molecules that may be absorbed into the circulating system. The absorbed molecules affect cellular processes, such as gene expression, differentiation, proliferation, and apoptosis [21].

### 3.1. Gut Microbiota and Short-Chain Fatty Acids

The microbiota that live in the human cecum and colon can produce carbohydrate-active enzymes that ferment non-digestible carbohydrates such as cellulose, xylans, and inulin to generate short-chain fatty acids (SCFAs) [22]; of these, propionate (C_3_), butyrate (C_4_), and acetate (C_2_) acid predominate in the ratio 1:1:3 [21]. These SCFAs are rapidly absorbed by epithelial cells in the gastrointestinal tract. The bacteria also synthesize and release isobutyrate and hexonate [23]. Of the SCFAs, acetate is released by most anaerobic bacteria, and propionate and butyrate by various subsets of gut microbiota. In addition, the bacteria of the human colon synthesize propionate from sugars via the succinate or propanediol pathway, or butyrate via glycolysis [24].

The gut microbiota has been estimated to produce 50–100 mmol/L/day of SCFAs, and that up to 95% of this serves as an energy source for colonocytes [25]. Particular SCFAs play several roles in humans. Butyrate is involved in anti-inflammation and anticancer processes [26]. It influences tight-junction assembly and mucin synthesis, thus enhancing gut barrier function and attenuating bacterial translocation [26]. Butyrate and acetate are inhibitors of histone deacetylase [27], resulting in histone hyperacetylation, altered gene expression, anti-inflammatory properties, the induction of growth arrest and apoptosis [28]; thus, the microbiota can influence the proliferation and differentiation of human colonic epithelial cells, and their gene expression, via butyrate and acetate production. It is believed that these SCFAs could regulate 2% of the mammalian transcriptome [29]. The inhibition of histone deacetylase activity by SCFAs also stimulates non-histone proteins, such as nuclear factor-kappa B (NFкB), MyoD, a transcription factor that could reprogram fibroblasts into skeletal muscle cells, p53 and the nuclear factor of activated T cells (NFAT), which can modulate gene expression [30]. Butyrate also inhibits the expression of pro-inflammatory mediators, such as interleukin 1β (IL-β), IL-6, IL-12, tumor necrosis factor-α (TNF-α), nitric oxide (NO) and interferon-γ (IFN-γ), but increases the release of anti-inflammatory IL-10 [30], which stimulates the expression of intestinal epithelial cytoprotective shock proteins (HSPs) 25 and 72 [31]. In addition, elevated levels of butyrate and propionate also activate intestinal gluconeogenesis via the gut–brain neural circuit [32]. Generally speaking, the SFCAs released by colonic bacteria are known to play a wide range of functions in the human body, including as signaling molecules [15].

### 3.2. Gut Microbiota and Bile Acids

The gut microbiota is known to synthesize secondary bile acids (BAs), deoxycholic acid (DCA), ursedeoxycholic acid (UDCA) and lithocholic acid (LCA) from cholesterol derived from the human liver [33]. This process takes place in the distal small intestine and colon [34]. The synthesized primary bile acids are conjugated to glycine and taken up in the distal ileum for transport to the liver. These bile acids are deconjugated by bacteria in the distal ileum and colon, and are then further metabolized by gut microbiota into secondary bile acids [20], such as deoxycholic acid (DCA) and litocholic acid (LCA). Like SCFAs, bile acids play several roles in the human body. They are naturally occurring detergents that help in the absorption of dietary fats. They also act as signal molecules: BAs bind to cellular receptors such as the farnesoid X receptor (FXR) [35] and G protein-coupled receptors, such as G protein-coupled bile acid receptor (TGR5), also known as a membrane-type receptor for bile acids (M-BAR) [20]. The activation of FXR, by primary bile acids, and TGR by secondary bile acids [20] influences the glucose metabolism: TGR5 activation stimulates cyclic adenosine monophosphate (cAMP) synthesis, which activates protein kinase A (PKA); FXR activation impairs glucose homeostasis, whereas the activation of TGR promotes it [36]. Results obtained in animal studies indicate that the stimulation of TGR5 stimulates the secretion of glucagon-like peptide 1 (GLP-1) from L-cells, enhances glucose tolerance in obese mice [37], and in brown adipose tissue and muscle, increases thermogenesis, and protects against diet-induced obesity [38].

### 3.3. Gut Microbiota and Protective Functions

The human gut microbiota is involved in the construction of the mucus layer and the secretion of mucins in the intestine [39]; it also enhances the intestinal barrier, which protects against the colonization of pathogenic bacteria such as *Streptococcus pneumoniae* [40]. The bacteria of the microbiota influence the development of the immune system [41], as well as of the humoral and cellular immune system [42]. The gut microbiota releases metabolites that act as signals for the development of regulatory T-cells (Treg), T-helper type 1 and type 2 cells, and T-helper 17 cells [43]. In addition, the aforementioned SCFAs produced by gut bacteria influence the production of inflammatory mediators of macrophages; these are believed to be involved in a range of diseases, such as atherosclerosis and rheumatoid arthritis, as well as various neurodegenerative diseases [30]. Their course is influenced by the production and secretion of large amounts of TNF-α, IL-β, IL-6, chemokines and nitric oxide by the activated macrophages, as well as various arachidonic acid derivatives, such as thromboxane A_2_ and F1α. Butyrate decreases the release of TNF-α, IL-6, IL-12 and IFN-γ, while increasing the release of anti-inflammatory IL-10 [44]. These SCFAs act as anti-inflammatory mediators, decreasing the release of macrophage chemoattractant protein (MCP), and increasing the release of IL-10 and prostaglandin E [45].

Germ-free (GF) animals, i.e., those devoid of microbes, typically exhibit immune deficiency. They are also characterized by a lack of CD4^+^ T cells, as well as defective T and B cell function, poorly developed lymphoid tissues, and decreased antibody production. In these animals, immune function may be restored by colonization of the GI. Alternatively, treatment with *Bacteroides fragilis* or its capsular antigen PSA (polysaccharide A) induces the proliferation of CD4^+^ T cells, and restores the development of lymphoid-containing spleen white pulp [46].

Lactobacilli regulate dendritic cell function, and influence the Th1/Th2/Th3 cytokine balance at the intestinal mucosa [47], as well as the activation of natural killer (NK) cells [48]. Gram-negative bacteria secrete peptidoglycan via NOD-like receptor (NOD1), which induces the formation of isolated lymphoid follicles (ILF) [49]. ILFs are organized lymphoid structures in the small intestine in humans and in mice. In mice, they contain a polyclonal population of B lymphocytes [50].

Certain strains of Clostridia (Cluster IV, XIVa, and XVIII) and Bacteroidetes also play a protective role in the body. These bacteria enhance the abundance of intestinal CD4^+^Foxp3^+^ regulatory T cells, which play several immunological functions, such as maintaining tolerance to commensal bacteria, suppressing the aggressive immune responses to auto- and bacteria antigens, and inducing epithelial wound repair [51]. By releasing products that enhance neutrophil function, the human gut bacteria influence the development of the host innate immune system [52].

### 3.4. Gut Microbiota and Neurotransmitters and Neuropeptides

Several bacteria from the genera *Lactobacillus*, *Bifidobacterium*, *Escherichia* and *Enterococcus* synthesize and release neurotransmitters and neuropeptides. The main inhibitory neurotransmitter in the central nervous system (CNS) is gamma-aminobutyric acid (GABA), and disturbances in the GABA system can result in chronic disorders, such as anxiety and depression. GABA is produced from glutamate by various bacteria, such as *Lactobacillus* and *Bifidobacterium* [53]. *Lacticaseibacillus rhamnosus* can thus modulate the expression of GABA receptors in specific regions of the brain, and possibly play a role in the therapy of depression and anxiety [54]. Serotonin, a monoamine neurotransmitter, is also involved in the regulation of several brain functions and the modulation of various processes, including motor function, mood, sleep, pain, aggression, and sexual behavior [55]. Many neuropsychiatric disorders, including anxiety and depression, are associated with disturbances in the serotogenic system [56]. While approximately 90% of serotonin is synthesized and released by enterochromaffin cells, it can also be produced by bacteria from the genera *Escherichia*, *Streptococcus* and *Enterococcus* [57]. Gut microbiota can also enhance serotonin synthesis by activating SCFAs on enterochromaffin cells [58].

The human gut microbiota also produces other bioactive signaling molecules. *Escherichia* spp., *Bacillus* spp., and *Serratia* spp. produce dopamine, *E. coli*, *Bacillus* spp. and *Saccharomyces* spp. synthesize norepinephrine, while *Lactobacillus* spp. produce acetylcholine and glutamate [59,60].

The CNS is a site of high expression of neurotrophin, a brain-derived neurotrophic factor (BDNF) which regulates multiple aspects of cognitive and emotional behavior [61]. This BDNF also influences neural survival and differentiation, and prevents neuronal damage and death, and decreased levels of BDNF have been observed in germ-free (GF) mice [62], and in the brains and serum of patients with Alzheimer’s disease [63]. Pretreatment of animal models with a probiotic such as *Bifidobacterium longum* normalizes the BDNF levels [62].

## 4. Modulators of Gut Microbiota Composition

The composition of the gut microbiota depends on several factors, such as the method of delivery, diet while newborn and later in life, use of pharmaceuticals, probiotics and prebiotics, and intestinal microflora transplantation. These are also influenced by demographic factors such as age, sex, and ethnicity [64].

### 4.1. Prenatal Factors

Prenatal factors affect human gut microbiota. Microbes have been detected in the placenta [65], amniotic fluid [66], fetal membrane [67], umbilical cord blood [68], and meconium [69]. For example, bacteria belonging to the genera *Enterococcus*, *Streptococcus*, *Staphylococcus*, and *Propionibacterium* have been isolated from umbilical cord blood and from the meconium samples [68,70], and the DNA of *Bifidobacterium* and *Lactobacillus* has been isolated from placental samples [65]. Studies on *Enterococcus faecium* in animals suggest that gut microbiota may be translocated from mother to fetus via the bloodstream [70]; as such, it has been suggested that, in the case of *Firmicutes*, *Tenericutes*, *Proteobacteria*, *Bacteroides*, and *Fusobacteria*, maternal oral bacteria may be translocated to the fetus via the bloodstream [71,72]. Therefore, unlike previously believed, bacteria are translocated from mother to fetus in healthy pregnancies [70], and fetuses are not sterile.

### 4.2. Method of Delivery

In the case of vaginal delivery, newborns develop microbiota within 20 min of birth from the maternal vaginal or fecal microbiota. As such, the intestinal microbiota of newborns are similar to the vaginal microbiota of their mothers [73], with the most abundant bacteria in the gut microbiota of infants being *Lactobacillus*, *Prevotella*, and *Sneathia* [74]. In contrast, newborns delivered by Cesarean section typically present different gut microbiota, these being more characteristic of the microbiota of the skin from the hand that touches them after birth [75]; in these newborns, the most abundant bacteria are *Staphylococcus*, *Corynebacterium*, and *Propionibacterium* [74]. However, a few days after delivery, the composition of the gut microbiota shifts to being dominated by Proteobacteria and Actinobacteria [7].

### 4.3. Method of Feeding

The gut microbiota in breast-fed infants is dominated by *Bifidobacterium* [76] and *Ruminococcus* [77]. This microbiota presents significantly lower contributions of *Escherichia coli*, *Clostridium difficile*, *Bacteroides fragilis*, and *Lactobacillus* spp. than those of formula-fed bottle-fed infants [78]. In bottle-fed infants, the composition of gut microbiota includes enterobacterial genera, such as *Streptococcus*, *Bacteroides*, *Clostridium*, *Bifidobacterium*, and *Atopobium* [79]. The introduction of solid food results in the formation of a more complex and stable microbiota, similar to adult gut microbiota [80].

### 4.4. Age

As a child gets older, the intestinal microbiota gradually stabilizes. After one year of age, the composition of the gut microbiota resembles that of a young adult [73]. Later, at approximately 2.5 years of age, it becomes dominated by Firmicutes and Bacteroidetes [7], and finally resembles adult microbiota at around the age of seven [74]. At this age, about 90% of the bacterial composition comprises Firmicutes and Bacteroidetes, with the remaining 10% being made up of Terricutes, Cyanobacteria, and Proteobacteria [74]. In addition to the aging process, the composition of human gut microbiota may also be influenced by a range of factors, including pharmaceutical use, swallowing difficulties, and digestive problems. The low-grade systemic inflammation observed in the aging process stimulates the growth of pathobionts; as such, the microbiota of the elderly tend to demonstrate lower biodiversity [81]. The number of anaerobic bacteria remains stable, however, the proportion of facultative anaerobes increases. This gut microbiota is dominated by *Bacteroides* and Firmicutes, with Actinobacteria and Proteobacteria representing a smaller fraction [81]. Interestingly, gut microbiota composition in elderly people has been found to vary depending on nationality: while German, Austrian and Finnish seniors demonstrate an increased proportion of Bacteroidetes, Italian seniors do not indicate this [81].

### 4.5. Diet

One of the important factors influencing the composition of the gut microbiota is diet [82,83,84]. The modern diet consumed in industrialized countries tends to be rich in fat and saturated fatty acids, and higher in carbohydrates than that of preagricultural people [85,86]; this diet is accompanied by lower levels of dietary fiber and products with a low glycemic index [87]. European children tend to prefer this “Western diet”, which is low in dietary fiber, but rich in animal protein and saturated fatty acids. Not surprisingly, compared to that of children in rural Africa, the gut microbiota of European children tends to be depleted in Bacteroides and bacteria from the genus *Xylanibacter*, but enriched in *Enterobacteriaceae* (*Shigella* and *Escherichia*), Firmicutes (*Faecalibacterium* and *Acetitomaculum*), and Gram-negative bacteria. The gut microbiota of the African children tends to contain a greater proportion of Bacteroidetes and Gram-positive bacteria, a higher ratio of Bacteroidetes:Firmicutes, and a greater abundance of *Prevotella* and *Xylanibacter*. The consumption of a Western diet also results in the loss of several bacterial species and a reduction in microbial diversity and stability [88], which are indicators of unhealthy microbiota [89]. The children from an African village in Burkina Faso were found to consume a low-fat but plant-rich diet [19]. The gut microbiota of the Hadza, a community of human hunter-gathers from Tanzania, was found to be characterized by much greater microbial richness and diversity than those of urban residents in Italy. The gut microbiota of the Hadza contained bacteria from the phylum Spirochaetes, the genera *Anaerophaga*, *Sphingobacteriales*, *Ruminobacter* and *Treponema*, and from family *Veillonellaceae*, whereas these taxa were absent from the Italian subjects; conversely, the microbiota of the Italian subjects over-expressed the phyla Actinobacteria and Firmicutes, as well as the genera *Bifidobacterium*, *Bacteroides*, *Alistipes* and *Blautia*, compared with the Hadza [88].

Other diets have also been found to influence the composition of human gut microbiota. A diet rich in plant-derived carbohydrates, digestible and non-digestible fiber, increases the proportions of *Actinobacteria*, *Bacteroidetes* (*Prevotella*, *Xylanibacterium*), *Proteobacteria*, *Bifidobacteria*, *Lactobacilli*, *Ruminococcus*, *Eubacterium rectale*, *Blautia*, *Streptococcus*, and *Bifidobacteria*. It also increases the diversity of the microbiota and Firmicutes:Bacteroides ratio, but decreases shares of *Firmicutes*, *Bacteroides*, *Clostridium* spp. *Enterococcus*, *Roseburia*, *Bacteroidaceae*, and *Eubacterium*. A vegetarian or vegan diet increases the proportion of *Bacteroides thetaiotaomicron*, *Clostridium clostridioforme*, *Faecalibacterium prausnitzii*, *Bifidobacterium*, and *Lactobacillus*, but decreases that of *Acteroides* spp., *Enterobacteriaceae*, *Escherichia coli*, *Bacteroides fragilis*, *Clostridium perfringens*, and *Clostridium* cluster XIVa. The Mediterranean diet, often regarded as the best diet, characterized by high amounts of fruits, grains, vegetables and monounsaturated fats, is associated with an increased share of *Bacteroidetes*, *Clostridium*, *Prevotella* and *Firmicutes*, and a lower share of *Bacillaceae* and *Proteobacteria*. Finally, a high-protein diet increases the shares of the Bacteroides enterotype, *Alistipes*, and *Bilophila*, and decreases that of Firmicutes, such as *Ruminococcus bromii*, *Roseburia*, and *Eubacterium rectale* [15,90,91,92].

### 4.6. Probiotics

Probiotics, treated as dietary supplements, are defined as “live microorganisms which when administered in adequate amounts, confer a health benefit on the host” [92]. Most bacterial probiotics currently sold are lactic acid bacteria *Ligilactobacillus salivarius*, *Lacticaseibacillus paracasei*, *Limosilactobacillus reuteri*, *Lactiplantibacillus plantarum*, and *Lactobacillus gasseri*) and *Bifidobacterium*, such as *B. lactis* [2]. Probiotics are involved in several human and animal functions, such as immune and metabolic processes, protection against diseases, and anti-tumorigenic effects [93]; they are also believed to influence the normalization of brain functions [59].

Animal studies indicate that probiotic use influences body weight. The administration of *Limosilactobacillus ingluviei* increases body weight [94], *L. plantarum* reduces adipocyte size in mice [95], and *L. paracasei* reduces fat accumulation [96]. Administration of *Lacticaseibacillus casei/paracasei*, *L. plantarum*, and *L. gasseri* demonstrated an anti-obesity effect [97]. Probiotic VSL#3, which contains eight different strains of bacteria belonging to the genera *Bifidobacterium* and *Lactobacillus*, stimulates the secretion of GLP-1, a hunger-reducing hormone [98]. Probiotics have also been found to influence mood. *B. longum* and *Lacticaseibacillus rhamnosus* administration in animals normalizes anxiety-like behavior caused by the parasite *Trichuris muris* [54], and in humans, *Lactobacillus* and *Bifidobacterium* treatment has been found to decrease the symptoms of anxiety [99]. Two-week administration of *Lactobacillus helveticus* and *B. longum* also alleviated anxiety and depressive symptoms in healthy volunteers [100]. They are also believed to influence the human brain, especially regions that control processes associated with emotion and sensation [101]; they have been found to decrease the levels of BDNF and modulate the levels of GABA [54], as well as a range of other functions [15]. It was found that probiotic treatment decreases the levels of C-reactive protein, total cholesterol, LDL-cholesterol, and plasma triglycerides, and increases those of SCFAs, IL-10, IGA, HDL-cholesterol; administration has also been associated with insulin sensitivity [102,103,104]. Administration of probiotics increases the counts of beneficial gut bacteria, such as *Bifidobacteria* and *Lactobacilli*, and significantly reduces those of enteropathogens, such as *E. coli* and *H. pylori* [105,106].

### 4.7. Prebiotics

A prebiotic is “a nondigestible food ingredient that beneficially affects the host by selectively stimulating the growth and activity of one or a limited number of bacteria in the colon, and thus improving host health” [107]. Prebiotics are not hydrolyzed by the enzymes of the human intestine, but are selectively fermented by colonic bacteria. Several compounds may serve as prebiotics, such as inulin, oligosaccharides, and dietary polyphenols. *Inulin* is the general term used to describe β(2–1) linear fructans with variable degrees of polymerization. Colonic bacteria ferment inulin to SCFAs and gases. Various oligosaccharides, such as fructooligosaccharides (FOSs), galactooligosaccharides (GOSs), soybean oligosaccharides (SBOSs) may act as probiotics [108], and vegetables such as Jerusalem artichokes and chicory, which contain inulin, are regarded as prebiotic-rich foods. Inulin increases the counts of beneficial bacteria, such as *Bifidobacteria* and *Lactobacilli* [109,110], and decreases the counts of enterococci [111], facultative anaerobes [112], and *Bacteroides* [109]. In obese women, inulin was found to increase the count of *Bifidobacterium* and *Faecalibacterium prausnitzii*, and decrease those of *Bacteroides* and *Propionibacterium* [113]. Uptake of Yacon syrup, which contains FOS or GOS, was found to decrease the body weight, BMI, and waist circumference in obese adults, as well as serum LDL-cholesterol and glucose levels [114]. Several investigations have found prebiotics to have a beneficial role in human metabolic disorders [15].

Prebiotics have also been found to influence brain functions. Arabinoxylan, a plant polysaccharide, stimulates the growth of beneficial bacteria known to produce butyrate, such as *Roseburia intestinalis*, *Eubacterium rectale* and *Anaerostipes caccae* [115]. It has also been proposed that plant polyphenols may also have prebiotic properties; these compounds are plentiful in fruits, nuts, seeds, and vegetables, as well as food products and beverages such as chocolate, coffee, red wine, and soymilk. Polyphenols increase the counts of *Bifidobacteria* and *Lactobacilli*, and decrease those of *Bacteroides*, *Clostridium*, *Salmonella typhimurium* and *Staphylococcus aureus* [116].

### 4.8. Pharmaceutical Use

Recent years have seen an increase in antibiotic therapy. Antibiotics significantly influence the composition of human gut microbiota, with some taxa not recovering for several months after treatment [117]: while the human intestinal microbiota can take four weeks to recover from five-day antibiotic treatment, some taxa may need six months to return to levels before therapy. Antibiotic therapy mainly reduces the diversity and/or abundance of Bacteroides, and decreases the evenness of the community [118]. Treatment with ampicillin and gentamycin within 48 h of birth in a mother has been associated with higher counts of Proteobacteria in the infant gut microbiota, and lower counts of Actinobacteria and the genus *Lactobacillus*, compared to those of untreated mothers. After eight weeks, the recovery was still incomplete, and the level of *Proteobacteria* remained elevated [119]. Several other antibiotics have been found to influence the infant gut microbiota. For example, administration of cephalexin reduces the levels of *Bifidobacterium* and increases those of *Enterococcus* and *Enterobacteriaceae* [120], and six-week intravenous treatment with vancomycin was found to change the level of *Lactobacillus* [121]. The use of antibiotic therapy in newborns may cause the overgrowth of *Clostridium difficile*, resulting in antibiotic-associated diarrhea [122]. Several of these observations have been confirmed in animal studies [123]. Interestingly, the use of antibiotics in animals has been found to increase the gastrointestinal abundance of *Candida albicans* [124].

Metformin, widely used in patients with type 2 diabetes mellitus, also influences the human gut microbiota. The levels of metformin in serum correlate positively with *Escherichia* counts, and negatively with Intestinibacter [125,126]. It has also been suggested that metformin may have beneficial effects by increasing the levels of butyrate and propionate producers [126].

### 4.9. Intestinal Microflora Transplantation

Intestinal microflora transplantation (IMT) [2], also known as fecal microbial transplantation (FMT) [127], was used more than a hundred years ago as a traditional Chinese medicine to treat diarrhea [128]. The technique has also been used in contemporary medicine, in patients presenting pseudomembranous colitis after antibiotic treatment of *Clostridium difficile*. Since then, IMT has been extended to treat other diseases, including those of the GI tract [7]. The aim is restoring normal intestinal microbial balance [129]. Fecal microbiota obtained from the filtrate stool of a “healthy” donor is transplanted to a recipient with a disturbed or altered microbiota. The method of introduction of microbiota depends on the disease. The sample is introduced by endoscopy into the large intestine in the case of *Clostridium difficile* infection, and into the duodenum in the case of metabolic syndrome [2]. This technique has frequently been used to treated *Clostridium difficile* infection, and has been found to be highly effective. These patients demonstrate decreased levels of Bacteroidetes and Firmicutes; however, 14 days after transplantation, the microbiota was dominated by *Bacteroides* spp. [130]. IMT has also been proposed as a treatment for inflammatory bowel disease [131], metabolic syndrome [132], type 2 diabetes mellitus [133], autism spectrum disorder and mood disorders [134], as well as Parkinson’s disease [135], Alzheimer’s disease, and several others [136,137]. Various next-generation therapies have also been proposed in which synthetic microbial composition in place of the microbial composition of healthy donors is used [138]. However, FMT is generally not well accepted by patients.

## 5. Metabolic Endotoxemia

Metabolic endotoxemia, first described in mice, is the term given to increased levels of lipopolisaccharides (LPS) in blood plasma [139]. LPS is a cell wall component derived from the lysis of Gram-negative bacteria in the intestine. After an absorption by enterocytes, LPS is carried in plasma and bound to chylomicrons [140]. In humans, the overgrowth of Gram-negative bacteria is closely associated with the consumption of a high-fat diet.

Toll-like receptors (TLRs) play a crucial role in the innate immune system, and the combination of LPS with CD14 can act as a ligand for TLR-4. LPS binds to the CD14/TLR-4 receptor in macrophages, triggering an inflammatory cascade [139]. In healthy humans, metabolic endotoxemia increases the levels of adipose TNF-α and IL-6, thus promoting insulin resistance [141]. The postprandial increase of plasma LPS observed after a high-fat meal increases the expression of NF-кB and a suppressor of cytokine signaling-3 (SOCS-3), which are involved in insulin resistance; no such changes are observed in the case of a diet rich in fiber and fruits [142]. Increased levels of plasma LPS stimulate the CD14/TLR-4 complex, which stimulates the TLR-2 mediated inflammatory response, thus increasing the secretion of pro-inflammatory cytokines by adipose tissue [143]. Animal studies based on mice have found that a high-fat diet changes the composition of gut microbiota; more specifically, a reduction in the abundance of *Bifidobacteria* spp. and *Eubacteria* spp. accompanied by a 2- to 3-fold increase in LPS level [144].

## 6. Dysbiosis

The term *dysbiosis* is defined as an “increase in the population of gut bacteria with pathogenic traits, which occasionally causes diseases” [145]. Hence, dysbiosis results in a change in the composition of the gut microbiota, characterized by a decrease in the share of symbionts and commensals, and/or an increase in the share of pathobionts. As mentioned earlier, several factors may change the balance and induce dysbiosis, which may directly or indirectly influence the course of several diseases [146,147,148,149,150]. In addition, fungal dysbiosis may also play a role in several diseases [14,151].

## 7. Human Microbiota Dysbiosis and Cancers

Carcinogenesis progresses through three stages: initiation, progression, and metastasis. The tumor requires a particular environment, known as the tumor microenvironment (TME), to grow and thrive. The TME promotes tumor transformation, growth and invasion, protects it from host immunity, and enhances resistance to therapeutics [152,153].

The development of cancer may be caused by several factors. Certain bacteria, viruses, and fungi may influence cellular dysplasia and carcinogenesis, while inflammation can also promote tumorigenesis and proliferation [154]. In most cases, carcinogenesis occurs secondary to the creation of a local chronic inflammation state. Some bacteria, such as *Helicobacter pylori*, may directly contribute to the process of tumorigenesis by influencing the intracellular signaling pathways involved in the regulation of cell growth and proliferation [155]. Examples of such oncogenic bacteria include *Salmonella typhi* and *Helicobacter* spp. in biliary cancer [156], and *H. pylori* in gastric cancer [155]. In addition, viruses such as hepatitis C virus (HCV) and hepatitis B virus (HBV) are known to support the development of hepatocellular carcinoma [157]. Approximately 16% of human cancers are caused by infectious factors or infection-associated chronic inflammation [158].

The process of tumorigenesis may involve various mechanisms. Some bacterial species, such as *Bacteroides fragilis*, stimulate inflammation, and the induction of pro-inflammatory toxins may promote carcinogenesis [159]; *Helicobacter hepaticus* stimulates the development of colon cancer in animal models, increasing the production of reactive oxygen species [160]; *Fusobacterium nucleatum* may alter signaling pathways and inhibit host antitumor immune function [161]; *Escherichia coli* produces genotoxic metabolites, such as colibactin, and *Campylobacter jejuni* produces a cytolethal toxin, known to induce carcinogenesis in animal models [162]. *Fusobacterium nucleatum* produces a FadA adhesion complex (FadAc), which activates the β-catenin-Wnt signaling pathway, influencing oncogene transcription in human colon cancer cell lines [163]. *F. nucleatum* has also been found to influence the development and progression of colon adenomas and colon cancers [164,165], and other bacterial species are believed to employ other mechanisms [166,167,168].

### 7.1. Human Gut Microbiota and Cancers of Digestive System

Several cancers of the digestive system are caused by the dysbiosis of gut microbiota. These bacteria are known to be involved in the initiation, progression, and metastasis of cancers in various organs of the human digestive system.

#### 7.1.1. Oral Cavity Cancers

Oral cavity cancer, primarily oral squamous cell carcinoma (OSCC), arises from oral mucosa. It is the fourteenth most prevalent malignancy worldwide [169]. Approximately 90% of oral cavity cancers are squamous cell carcinoma.

The healthy oral cavity is home to 772 bacterial species, belonging to 185 genera and 12 phyla. The most abundant phyla, constituting 96% of oral bacteria, are the Firmicutes, Actinobacteria, Fusobacteria, Bacteroides and Spirochaetes. Lower levels of Chloroflexi, Chlamydiae, Saccharibacteria and Gracilibacteria are also observed [170,171]. In a healthy oral cavity, the predominant bacteria include 12 Gram-positive genera, including *Abiotrophia*, *Peptostreptococcus*, *Actinomyces*, *Bifidobacterium* and *Lactobacillus* and 15 Gram-negative genera, including *Neisseria*, *Veillonella*, *Capnocytophaga*, *Fusobacterium* and *Prevotella* [172]. Other nonbacterial organisms have also been recorded, including the protozoa *Entamoeba gingivalis* and *Trichomonas tenax*, and 85 fungal genera, such as *Candida*, *Aureobasidium*, *Saccharomycetales*, *Aspergillus*, *Fusarium*, and *Cryptococcus* [173]. Taxonomic diversity within samples is known as *α-diversity*, and diversity between samples as *β-diversity* [174].

Several environmental factors, including changes in the composition of microbiota, may be involved in the development of oral squamous cell carcinoma (OSCC). Several authors have suggested potential associations between oral dysbiosis and oral cancer [175]. In patients with OSCC from Sri Lanka, cancer tissue samples demonstrated lower species richness and diversity, characterized by the overexpression of certain genera, such as *Capnocytophaga*, *Pseudomonas* and *Atopobium*, and species such as *Campylobacter concisus*, *Prevotella salivae*, *P. loeschii*, and *Fusobacterium oral* taxon 204, as compared to control patients with fibroepithelial polyps. The OSCC tissue demonstrated a predominance of proinflammatory bacterial attributes, such as the biosynthesis of LPS and peptides [176]. Of note, the increased expression of genes involved in bacterial chemotaxis, flagellar assembly and biosynthesis is also associated with pathological processes [177]. Another investigation revealed drastic changes in the composition of oral microbiota in patients with OSCC. The cancer samples demonstrated significantly higher bacterial diversity than healthy controls, with the cancer samples demonstrating significantly higher levels of *Fusobacterium*, *Peptostreptococcus*, *Dialister*, *Filifactor*, *Peptococcus*, *Catonella*, and *Parvimonas*. Significantly higher diversity and richness in tumor sites have also been noted in other investigations of OSCC [177]. Cancer tissues were found to be significantly enriched in six families: *Prevotellaceae*, *Fusobacteriaceae*, *Flavobacteriaceae*, *Lachnospiraceae*, *Peptostreptococcaceae* and *Campylobacteriaceae*, and 13 genera, including *Fusobacterium*, *Allomevotella* and *Porphyromonas*. At the species level, *Fusobacterium nucleatum*, *Prevotella intermedia*, *Aggregatibacter seqnis*, *Capnocytophaga leadbetteri*, *Peptostreptococcus stomatis*, among others, were all found to be significantly increased in patients with OSCC [177]. Other studies have noted an association between dysbiosis in the oral microbiota and the development of OSCC [178,179,180]. It has been proposed that some oral taxa, especially *Porphyromonas gingivalis* and *Fusobacterium nucleatum*, may have carcinogenic potential [181,182], and some aerobic bacteria, such as *Parvimonas*, could also be linked to tumorigenesis [183]. It has also been suggested that oral and gut microbiota may serve as potential biomarkers in cancer [184,185], with three oral bacterial species being good candidates as diagnostic indicators of OSCC: *Capnocytophaga gingivalis*, *Prevotella melaninogenica*, and *Streptococcus mitis* [179]; *Peptostreptococcus* spp. and *Porphyromonas gingivalis* have also been proposed [186]. Interestingly, *Salmonella typhimurium* and *Clostridium* spp. may be used for targeted strategies as a potential vectors to treat cancer [182].

Gingival carcinoma is the third most prevalent oral cancer, with less than 10% morbidity. Higher levels of LPS biosynthesis were observed in the subgingival plaque of gingival squamous cell carcinoma (GSCC). The authors suggest that high levels of *Atopobium* in saliva and LPS synthesis have the potential risk of suffering in individuals with periodontitis [187].

Three mechanisms have been proposed for the influence of oral microbiota on the development of cancer [188]. The first involved stimulation of chronic inflammation by bacteria such as *Porphyromonas gingivalis*, *Fusobacterium nucleatum*, *Streptococcus anginosus*, and *Prevotella* spp. [186,189]. Oral epithelial cells produce pro-inflammatory cytokines and chemokines, including IL-1β, IL-6, IL-8, IL-23, TNF-α, and metalloproteinases, such as matrix metalloproteinase-8 (MMP-8) and MMP-9 [190]; these inflammatory mediators may induce or facilitate cell proliferation, mutagenesis, oncogene activation and angiogenesis [191]. The resulting chronic inflammation may cause immunosuppression, influencing the development of OSCC [192]. Alternatively, oral bacteria may affect cell proliferation, cytoskeletal rearrangement and NF-кB activation, and inhibit apoptosis. *Porphyromonas gingivalis* increases the level of cyclin A, diminishes the level and activity of p53, and activates the PI3-K pathway, inducing the proliferation of gingival epithelial cells [193]. Interestingly, the PI3-K/AKT and JAK/Stat3 signaling pathways are known to inhibit gingival epithelial cell apoptosis [194]. *P. gingivalis* may also cause immunosuppression [189]. Finally, bacteria such as *Porphyromonas gingivalis*, *Prevotella intermedia*, *Aggregatibacter actinomycetemcomitans*, and *Fusobacterium nucleatum* are known to produce various substances that may have carcinogenic properties, such as reactive oxygen species, reactive nitrogen species, organic acids, and various volatile sulfur compounds (VSC) such as hydrogen sulfide, methyl mercaptan and dimethyl sulfide [186,188]. Other oral microbiota, such as *Neisseria* sp. and *Candida* sp., produce acetylaldehyde and N-nitrosamine compounds, which may also be carcinogenic [195]. Colonization with fungal species also may be a risk factor in OSCC. Patients with OSCC demonstrated significantly higher levels of *Candida albicans* than healthy controls [196], and the *C. albicans* isolated from patients with OSCC were found to be genotypically different from those isolated from healthy subjects [197]. It has been suggested that *C. albicans* isolated from cancer tissue may demonstrate greater acetylaldehyde production and nitrosylating activity [198]. *C. albicans* was found to induce neoplastic mucosal changes in an animal model of chemically induced carcinogenesis [199]. Other studies have indicated an increased abundance of fungal species, such as *Candida etchellsii* and *Hanaella luteola*, in patients with OSCC [195]. One particularly interesting observation is that *Candida* colonization appears to be associated with increased levels of *F. nucleatum* [200].

#### 7.1.2. Esophageal Cancer

Esophageal cancer (EC) is the eighth most common cancer worldwide, and the sixth most common cause of cancer-related death [201]. There are two major histological types of esophageal cancers: esophageal adenocarcinoma (EAC) and esophageal squamous cell carcinoma (ESCC) [202]. EAC localizes in the distal esophagus, whereas ESCC localizes in the middle thoracic esophagus. In developed countries, the most common type is EAC, whereas ESCC predominates in developing countries [203]. EAC develops from the precursor lesion Barrett’s esophagus (BE), a characteristic metaplastic change of the mucosa of the distal esophagus [204]. Only 1 in 860 patients with BE will develop EAC [205]; however, more than 85% of patients with newly diagnosed EAC have no history of BE [206].

Healthy esophagus tissue has been found to include bacteria from 97 species belonging to six phyla: Firmicutes, Bacteroides, Actinobacteria, Proteobacteria, Fusobacteria, and TM7. Of these, the most common genera are *Streptococcus* (39%), *Prevotella* (17%), and *Veillonella* (14%) [207,208]. No difference in microbiota was found between benign and malignant tissues in surgically resected samples from patients with EAC and ESCC; in addition, the same bacteria were detected in normal and cancerous tissues [209]. More recent studies revealed a higher abundance of *Treponema denticola*, *Streptococcus mitis*, and *S. anginosus* in EAC samples compared to healthy controls; however, the pathological subtypes of the tumors were not specified [210]. Tissue samples from patients with EAC demonstrated decreased diversity [211], decreased levels of *Veillonella* and *Granulicatella*, and increased levels of *Limosilactobacillus fermentum*. It has been suggested that the presence of the genus *Neisseria* and the species *L. fermentum* may be associated with the risk of EAC [212]. In addition, chronic infection with *Helicobacter pylori* also appears to be inversely related to the risk of EAC [206,213]; however, the mechanism by which *H. pylori* affects carcinogenesis in the lower esophagus remains unclear [213]. There is also a need to better clarify the associations between the microbiome and EAC [214].

Little evidence exists regarding the microbiome in patients with ESCC [214]. Patients with esophageal dysplasia, a precursor of ESCC, demonstrated significantly lower numbers of microbial genera in the upper gastrointestinal tract, as well as the presence of *Porphyromonas gingivalis*, which was not observed in healthy control subjects [215]. Positive correlations have also been noted between the presence of *P. gingivlis* and cancer cell differentiation, metastasis, and poor clinical outcome [216]. Changes in the composition of bacterial microbiota in the saliva is associated with a higher risk of ESCC; patients with ESCC demonstrated a decreased abundance of the genera *Lautropia*, *Bulleidia*, *Catonella*, *Corynebacterium*, *Moryella*, *Peptococcus*, and *Cardiobacterium* as compared to healthy controls [217]. It has also been noted that patients with ESCC displayed a higher proportion of Firmicutes, lower share of Gammaproteobacteria, and higher proportion of *Bacilli* than in healthy controls. In these patients, α-diversity and richness were significantly lower than in healthy subjects, whereas β-diversity was higher [202]. It has also been suggested that a key role in the development of ESCC may be played by *Fusobacterium nucleatum*, a species that primarily inhabits the oral cavity, and has been detected in colon cancer tissue. This bacterium was detected in cancer tissue in about 23% of patients who underwent surgical resection of esophageal cancer. Its presence in cancer tissue is also associated with significantly shorter survival time [218].

#### 7.1.3. Gastric Cancer

Gastric cancer (GC) is the fifth most common cancer in the world, and the third leading cause of cancer-associated death worldwide [201]. Although GC is associated with various etiologies, *Helicobacter pylori* infection is the most well-established risk factor for its development [219]. Although *H. pylori* is known to induce chronic gastritis, and is associated with more than 90% of GC cases [220], GC is only observed in 1–3% of patients infected with *H. pylori* [221,222]. Nevertheless, *H. pylori* has been recognized by the WHO as a “definite carcinogen” [220]. The carcinogenesis of GC may involve other factors. The composition of gastric microbiome depends on gastric acidity [223], and due to its acidic environment, only *H. pylori* is able to colonize the human stomach. Such colonization causes chronic inflammation, which is believed to be the first step to GC. In the case of antral-predominant gastritis, bacteria stimulate the secretion of gastrin; although this causes further production of gastric acid and increases the risk of duodenal ulcers, it also protects patients against GC. In the case of corps-predominant gastritis, *H. pylori* suppresses the production of gastric acid through inflammatory mediators. This disturbance may cause the progressive loss of gastric glands, and leads to atrophic gastritis [224]. Corps-predominant gastritis may influence the development of carcinogenesis. Changes of gastric pH, due to reduced secretion of gastric acid, allow colonization by bacteria other than *H. pylori* from the oral cavity, upper respiratory tract, or the intestine; these bacteria are not normally detected in a normal stomach, because they cannot survive the hostile gastric environment. Changes of gastric pH may also be associated with the increased abundance of nitrosating species of bacteria in the stomach, as well as elevated nitride and N-nitrosamine levels [225].

A comparison of the gastric mucosal microbiota between *H. pylori*-negative and *H. pylori*-positive patients indicated that the *H. pylori*-positive gastritis patients demonstrated significantly lower bacterial richness. The *H. pylori*-negative patients also demonstrated increased levels of bacteria from the phyla Firmicutes, Fusobacteria, Bacteroidetes, and Actinobacteria; they also demonstrated higher counts of *Streptococcus* spp. and *Haemophilus influenzae* and, unlike the positive patients, they were found to display *Treponema* spp. The authors suggest that *Streptococcus* spp., *Haemophilus parainfluenzae*, and *Treponema* spp. may be potential pathogenic bacteria for the *H. pylori*-negative gastritis group. In addition, the relative abundance of *H. pylori* was found to be around 90% (α-diversity) in the *H. pylori*-positive group, while greater β-diversity was observed among *H. pylori*-negative patients [226]. Patients after gastrectomy for gastric cancer were also found to have higher species diversity and richness than healthy controls, with a higher abundance of aerobes, facultative anaerobes, and oral microbiota. Significantly higher levels of *Bifidobacterium nucleatum* were also observed, a species known to be related to colorectal cancer [227]. In addition, higher diversity and richness have been observed in cancerous tissue in comparison with non-cancer tissue [228]; the cancer samples were dominated by oral bacteria, such as *Peptostreptococcus*, *Streptococcus*, and *Fusobacterium*, and the non-cancerous controls by lactic acid-producing bacteria such as *Lactobacillus lactis* and *Levilactobacillus brevis*. In the cancerous samples, the predominant phylum was Proteobacteria, overall being found in 90% of cancerous samples, followed by Firmicutes, Bacteroidetes, Actinobacteria, and Fusobacteria. The non-tumor samples demonstrated higher levels of Proteobacteria than the cancerous samples, and lower levels of Firmicutes, Actinobacteria, and Fusobacteria [228].

Bacteria that can produce N-nitroso compounds, such as *E. coli*, *Lactobacillus*, *Nitrospirae*, *Clostridium*, *Veillonella*, *Haemophilus*, and *Staphylococcus*, have also been associated with GC [229]. In addition, *Streptococcus*, *Prevotella*, and *Neisseria* are treated as low risk factors for the development of GC, and it has been proposed that *H. pylori* changes the composition of the gastric microbiota, facilitating the colonization of stomach by oral bacteria [230]. This dysbiosis may influence the maintenance of the local microenvironment, stimulating the development and/or progression of GC [231,232,233,234]; however, the role of the gastric microbiome in the development of gastric cancer remains unclear [235]. Nevertheless, it has been suggested that *H. pylori* eradication therapy may be an effective way to prevent gastric cancer.

#### 7.1.4. Colorectal Cancer

Colorectal cancer (CRC) is one of the most commonly diagnosed malignances, and the third most prevalent cancer worldwide [163]. Its development has been closely associated with diet, suggesting that the gut microbiota may play a role in the development and progression of CRC [236]. A reasonable body of evidence exists to support this theory [237]. Higher levels of bacteria belonging to the group *Bacteroides-Prevotella* have been noted in stool samples of patients with CRC compared to healthy controls, and increased levels of *Enterococcus*, *Escherichia*, *Shigella*, *Klebsiella*, *Streptococcus*, and *Peptostreptococcus* were recorded in the luminal compartment of these patients [238]. These patients also demonstrated lower levels of butyrate producing bacteria belonging to the family *Lachnospiraceae*, as reported previously [239]. Several other studies have also reported higher proportions of *Fusobacterium*, *Parvimonas*, *Butyrivibrio*, *Gemella*, *Fusobacteria*, and *Akkermansia* in patients with CRC, and lower levels of *Ruminococcus*, *Bifidobacterium*, *Eubacteria*, and *Lachnospira* compared to healthy subjects [240,241,242].

The microbiome of CRC patients is often enriched in pro-inflammatory opportunistic pathogens and bacteria that influence the development of metabolic diseases, such as *Streptococcus bovis*, *Fusobacterium nucleatum*, *Escherichia coli*, *Bacteroides fragilis*, and *Enterococcus faecalis*, and lower levels of butyrate-producing bacteria, such as those of the *Roseburia*, *Clostridium*, *Faecalibacterium*, and *Bifidobacterium* genera [243,244,245]. High levels of *Bacteroides* increase the risk of colon polyps, whereas *Lactobacillus* and *Eubacterium* play a protective role [246]. In addition, higher levels of bacteria producing hydrogen sulfide and bile salts are also indicators of an increased risk of CRC development [247].

The stage of carcinogenesis has also been found to be associated with the composition of gut microbiota. In patients with adenoma, the intestinal mucosal surface demonstrated an increased amount of Firmicutes, Bacteroidetes, and Prevotella, in comparison with healthy controls [248]. However, resected tissues from patients with adenocarcinoma displayed higher levels of bacteria belonging to the phylum Bacteroides and lower levels of Firmicutes; in addition, an overabundance of Fusobacterium was found in samples of tumor as compared to healthy controls [249].

Several explanations have been proposed for the role of microbial dysbiosis in the development of CRC. It has been suggested, for example, that microbiota may have an epigenetic influence on host DNA expression, and that gut microbiota dysbiosis may promote carcinogenesis via epigenetic dysregulation based on gene methylation [237]. Animal studies revealed that germ-free mice that receive feces from patients with CRC showed higher numbers of hypermethylated genes in the colonic mucosa compared to those that receive feces from healthy controls. These mice also demonstrated a higher rate of colon epithelial renewal, and more precancerous lesions [237].

Alternatively, CRC patients have been found to demonstrate increased levels of *Fusobacterium nucleatum*. An *F. nucleatum* surface protein, FadA, binds to E-cadherin on epithelial cells, activates oncogenic pro-proliferative β-cadherin signaling [250], and stimulates the production of inflammatory cytokines such as IL-6, IL-8, and IL-18 [251]. This bacterium may also alter the function of tumor-infiltrating lymphocytes (TIL) and natural killer (NK) cells. It binds to the inhibitory immune receptor TIGIT (T cell immunoreceptor with Ig and ITIM domains), through another adhesion molecule, Fap 2 [161]. TIGIT is found as a protein on NK cells, and its activation enhances tumor killing [252], and the binding of bacteria to TIGIT blocks the anti-tumor activity of NK cells [161]. *F. nucleatum* can also activate the NF-кB pathway, thus inducing the expression of genes encoding pro-inflammatory cytokines [250,251]. An elevated level of NF-кB transcripts, observed in tumor tissue, decreases the levels of CD3-positive T-cells, suggesting that *F. nucleatum* obstructs the antitumor immune mechanism in cancer patients [253]. Indeed, in patients with stage III/IV of cancer, higher levels of *F. nucleatum* have been observed in tumor tissues than healthy tissues. A positive correlation has also been found between the levels of *F. nucleatum* and tumor invasion, and lymph node and distal metastasis [254].

*Bacteroides fragilis*, an enterotoxigenic bacteria (ETBF) which causes diarrhea and inflammation of the gastrointestinal tract, is also involved in colorectal carcinogenesis. Its presence is observed in biofilms coating human colorectal cancers and adenomas [255]. The *B. fragilis*-derived toxin (BFT) produced by these bacteria activates spermine oxidase produced by the host; its activation generates hydrogen peroxide and reactive oxygen species that damage the DNA in epithelial cells, promoting tumorigenesis [256]. The produced toxin also interacts with epithelial E-cadherin, disrupting the intracellular junction and activating β-catenin nuclear signaling, which induces cell proliferation [257]. These bacteria may induce the production of inflammatory cytokine IL-17, thus decreasing host anti-cancer immune responses and allowing cancer growth [258].

Higher levels of *Escherichia coli* have been observed in patients with CRC in comparison to healthy subjects [244]. This enterogenic bacterium expresses the genomic island polyketide synthase (pks^+^), which has been found to enhance tumorigenesis in preclinical models of CRC. *E. coli* pks^+^ produces the genotoxin colibactin [259] which alkylates DNA, causing double-strand breaks in DNA mammalian cells [260]. Colibactin also triggers premature and transmissible cellular senescence in the cells that initially survive DNA damage [261]. The influence of *E. coli* on the development of CRC was also confirmed in animal studies [259].

#### 7.1.5. Pancreatic Ductal Adenocarcinoma

Pancreatic ductal adenocarcinoma (PDAC) is a lethal and highly aggressive malignancy [262,263]. Growing evidence suggests that gut microbiota may play a role in PDAC susceptibility, initiation, and progression [264,265], as noted in animal studies [266]. The pancreas is generally considered to be free of microbes due to the presence of numerous proteases and a highly alkaline environment [267]; however, previous studies indicate increased bacterial abundance in cancerous pancreas tissues [268]. For example, patients with PDAC demonstrated 1000-fold greater levels of intrapancreatic bacteria in comparison with healthy pancreatic tissue [269]; the presence of *Fusobacterium* sp. in pancreatic tissues is correlated with poor pancreatic cancer prognosis [270], and the presence of *Gammaproteobacteria* in PDAC tissue inhibits the therapeutic effects of the anticancer drug, gemcitabine; it is believed that it is metabolized by the bacteria [271].

*H. pylori* is also believed to play a role in the initiation of pancreatic carcinogenesis [272]. Several subspecies of *Helicobacter* have been identified in the pancreas [273], and they have been found to produce a range of bacterial pathogenic compounds, such as ammonia, LPS, and inflammatory cytokines, which may damage the pancreas [274]. *H. pylori* can also dysregulate cellular processes by activating NF-кB and activator protein-1 (AP-1) and inducing mutations in *K-Ras*: these were observed in over 90% of investigated tissue samples [275]. The mutated cells are hyperstimulated by LPS, initiating the process of pancreatic carcinogenesis [276]. Infection of *H. pylori* also causes the activation of signal transducer and activator of transcription 3 (STAT3), resulting in the overexpression of anti-apoptotic and pro-proliferative proteins such as Bcl-xL, MCL-1, c-myc, and cyclin D1, which may promote cancer progression [277].

The risk of PDAC may also be increased by the dysbiosis of oral microbiota associated with periodontal disease and tooth loss [273,278]. It is suggested that the presence of such dysbiosis promotes the development of pancreatic cancer, rather than the other way round. For example, blood samples of patients with PDAC demonstrated significantly higher levels of antibodies against *Porphyromonas gingivalis*. This bacterium is a known periodontal pathogen which causes chronic periodontitis, and its greater abundance is associated with a twofold greater risk of cancer development [279]. In addition, the presence of *P. gingivalis* and *Aggregatibacter actinomycetemitans* in the oral cavity is believed to increase the risk of PDAC, whereas that of *Fucobacteria* and *Leptotricha* decreases it [280]. The main oral bacteria involved in the development of PDAC are *P. gingivalis*, *Fusobacterium*, *Neisseria elongata*, and *Streptococcus mitis* [273].

In addition, *P. gingivalis*, *Treponema denticola*, and *Tannerella forsythia*, known as “the red complex”, may also play a role in PDAC carcinogenesis. They are believed to be the key pathogens causing periodontitis. These bacteria secrete peptidyl-arginine deiminase (PAD) enzymes, which may cause point mutations in *p53* (tumor suppressor gene), and the oncogene *K-Ras* [281], the presence of which indicates a poor prognosis for patients with PDAC [282].

*Fusobacterium* spp. are also known to cause periodontal disease [283]. As this bacterium increases the production of reactive oxygen species and inflammatory cytokines and modulates the tumor microenvironment, it may be involved in the process of tumorigenesis [284]; in addition, the presence of *Fusobacterium* has been associated with a reduced risk of PDAC development [279,280]. Several other associations have also been found between oral microbiota and PDAC [285,286], which have also been confirmed in animal studies [266].

It has also been found that cancer patients with long-term survival (median survival 10.1 years) demonstrated higher α-diversity in the composition of tumor microbiome following surgical resection than those with short-term survival (median survival 1.6 years). The authors propose that α-diversity may be treated as a predictor of survival in patients with PDAC after surgical resection [286]. In addition, the patients with long-term survival demonstrated significantly higher levels of *Pseudoxanthoma*, *Saccharopolyspora*, and *Streptomyces* than those with short-term survival, suggesting that these three genera, and additional *Bacillus clausii* may predict long-term survival in PDAC patients [286].

The most common pancreatic cystic neoplasms (PCNs) are intraductal papillary mucinous neoplasms (IPMNs) [287]. These pancreatic cysts are characterized by malignant transformation to invasive carcinoma [288], from PCN, through low-grade dysplasia (LGD), high-grade dysplasia (HGD), to invasive carcinoma [287]. IPMN may be treated as an early indicator of pancreatic cancer [278]. Higher levels of *Fusobacterium nucleatum* and *Granulicatella adiacens* were found in samples obtained from patients with IPMNs compared to healthy controls, and higher levels of *Granulicatella*, *Serratia*, and *Fusobacterium* were observed in samples of HGD; lower levels of *Methylbacterium*, *Sphingomonas*, and *Propionibacterium* were also observed in the two groups [278]. Of note, these bacteria are oral microbiota.

#### 7.1.6. Hepatocellular Carcinoma and Cholangiocarcinoma

Although hepatocellular carcinoma (HCC) is the most common histological type of liver cancer [289], little research has been performed on the associations between gut microbiota and HCC in humans. Despite this, investigations of fecal samples of patients with HCC revealed higher levels of *Escherichia coli* in comparison with healthy controls [290,291], as well as decreased levels of *Bifidobacterium* and increased levels of *Bacteroides* and *Ruminococcus* [292]. A Chinese population of patients with early HCC demonstrated higher levels of the phylum Actinobacteria and 13 genera, including *Gemmiger*, *Parabacteroides*, *Prevotella*, *Alistipes*, *Phascolarctobacterium* and *Ruminococcus*, and lower levels of *Verrucomicroba*, *Klebsiella* and *Haemophilus* in comparison with healthy controls. Hence, patients with HCC demonstrated lower levels of butyrate-producing bacteria and higher levels of LPS-producing genera; they also displayed greater gut microbial diversity [293].

*Helicobacter* spp. has also been implicated in the development of HCC. It has been detected in the liver of patients with primary hepatic carcinoma, but not in healthy controls [294]; in addition, *H. hepaticus* was not detected in HCC patients with chronic hepatitis B or C [295]. *H. pylori*, *H. bilis*, *H. hepaticus*, and *H. ganmani* are specifically associated with cholangiocarcinoma, but not non-tumor diseases in the bile duct [296]. *Salmonella typhi* has also been associated with gallbladder and hepatobiliary carcinoma. HCC patients have also demonstrated changes in the composition of the tongue coating, and it has been suggested that the enrichment of tongue *Oribacterium* and *Fusobacterium* may be treated as a biomarker of HCC [297]. Further associations between the dysbiosis of gut microbiota and development of HCC have been described by other researchers [150,289,298,299,300].

Human gut microbiota may produce LPS, which can activate Toll-like receptor 4 (TLR4), causing the development of liver cancer [301]. Animal studies have found that lipopolysaccharide influences hepatic Kupffer cells, which produce the inflammatory cytokines TNF-α and IL-6. Cytokines activated by the LPS-TLR4-NF-кB signaling pathway can induce HCC by stimulating precancerous hepatocellular proliferation [302]. The development of hepatic cancer may also involve secondary bile acids, which are products of the bacterial metabolism of primary bile acids. These compounds influence the liver immune system [303,304], suppressing antitumor immunity, and promoting the progression of liver cancer. This suppression is believed to act through the prostaglandin E receptor on CD8 T cells [305].

### 7.2. The Human Gut Microbiota and Cancers of Urogenital System

The healthy urinary tract has been considered sterile, and the presence of bacteria was treated as some type of infection in the urinary tract. It is widely accepted that urine is hostile to the survival of microorganisms. The composition of the urinary microbiome differs between men and women. For example, female urine samples demonstrate a more heterogenous composition of bacterial composition, with *E. coli* being observed in 91% of healthy adult women, and only in 25% of men; in addition, the members of Actinobacteria (e.g., *Actinomyces*, *Arthobacter*) and Bacteroidetes (e.g., *Bacteroides*) are present in female urine samples, but not in male urine samples [306]. Further differences in the composition of urine microbiota between men and women have been given elsewhere [307].

Although little concrete data exist on the association between microbiome and kidney cancer; it has been proposed that some viral infections may influence the development of renal cancer. However, the evidence is controversial and sometimes contradictory.

#### 7.2.1. Human Microbiota and Bladder Cancer

Bladder cancer is the most prevalent malignancy of the urinary system [308,309], and is most commonly observed in men. However, although some associations have been suggested, no hard evidence exists to link dysbiosis of the urinary tract microbiota with bladder cancer [310].

Several investigations have indicated differences in the composition of urine microbiota between patients with bladder cancer and healthy subjects. For example, the urine samples of cancer patients demonstrated higher average number genera in comparison with healthy subjects, with the predominant genera being *Acinetobacter* and *Streptococcus*. The urinary samples of patients with bladder cancer demonstrate higher levels of *Actinomyces*, especially *A. europaeus*, as compared to those of healthy subjects [311]. While no significant differences in microbial diversity or overall microbiota composition were found between urine samples from male patients with bladder cancer and healthy controls, the former were found to demonstrate bacteria of the genus *Fusobacterium*, a possible pro-tumorigenic pathogen, while the latter were more abundant in the genera *Veillonella*, *Streptococcus*, and *Corynebacterium*. [308]. In the urine of cancer patients with high risk of recurrence and progression of bladder cancer, researchers observed the enrichment of *Herbaspirillum*, *Porphyrobacteria*, and *Bacteroides* [310]. Other studies have also described differences in the composition of urine microbiome between cancer patients and healthy subjects [312]. Interestingly, the condition of the bladder microbiota may prevent of the recurrence superficial bladder cancer, as is observed in the case of *Lactcaseibacillus casei* [313].

#### 7.2.2. Human Microbiota and Ovarian Cancer

Epithelial ovarian cancer (OC) is the second deadliest cancer of the female reproductive system [314]. One reason for this high mortality rate is the lack an effective screening method for early detection of OC; as such, more than 60% of cases are diagnosed at an advanced stage [315]. It has been proposed that the peritoneal inflammation caused by OC may be a possible mechanism for cancer metastasis [316,317]. While studies suggest that *Chlamydia trachomatis* and *Mycoplasma genitalium* may play a part in the development of OC [318], few studies have described the association between microbiota composition and ovarian cancer.

As human microbiota can alter the metabolism of estrogen, they may suppress or promote estrogen-derived cancers [319,320]. In addition, it is possible that OC influences the composition of local microbiota, thus altering its microbial environment. Investigations of peritoneal fluid from patients with advanced OC, III and IV stages revealed the presence of various bacteria that may be associated with disease pathogenesis. These included the *Rikenellaceae* (phylum Bacteroidetes), known as estrogen responsive, *Alphaproteobacteria* (phylum Proteobacteria), involved in vascular permeability, and *Akkermansia muciniphila* (phylum Verrucomicrobia), which show anti-inflammatory properties. The authors add a list of 18 microbial features, that appear to be closely associated with the pathology of OC, and suggest that the composition of peritoneal microbiota may be important for the diagnosis of OC [315]. Elsewhere, in samples of ovarian cancer found to demonstrate *Brucella* [321], *Mycoplasma*, in 59% of the ovarian cancer tissue samples [322], and *Chlamydia*, in 70% [323]. These bacteria were not detected in healthy subjects [323]. Other studies have noted the presence of *Streptococcus*, *Stphylococcus*, *Bacillus*, *Pediococcus*, *Chyseobacterium*, *Fusobacterium*, *Prevotella*, *Salmonella*, *Escherichia*, and *Treponema* in ovarian cancer samples [314]. Most importantly, these bacteria are also associated with other cancers.

Associations have been noted between the profile of the cervicovaginal microbiome, *BRCA1* mutations, and the risk of OC [324]. In addition, it was found that a microbiome in which *Lactobacillus crispatus*, *L. iners*, *L. gasseri*, and *L. jensenii* account for at least 50% of the bacterial species may protect against ovarian cancer. If *Lactobacillus* spp. accounted for less than 50% of all bacterial genera detected, this microbial composition may be involved in development of OC [324]. *Lactobacillus* spp. Also plays a protective role against vaginal infection and inflammation [325].

#### 7.2.3. Human Microbiota and Cervical Cancer

The risk of development of cervical cancer is increased by infection with human papillomavirus (HPV) [326]. Women infected with HPV have greater bacterial diversity [327], and are specifically abundant in *Lactobacillus gasseri* and *Gardnerella vaginalis*. A higher level of HPV was associated with a lower level of *Lactobacillus* and greater vaginal microbiome diversity [328]; in addition, the level of *Sneathia* was found to closely correlate with HPV infection [329]; furthermore, an elevated level of *L. gasseri* or *L. iners* correlated with rapid remission of HPV, and microbiota with low levels of *Lactobacillus* spp. And a high level of *Atopobium*, demonstrated slower HPV clearance [327]. It has also been proposed that *Chlamydia trachomatis* may change the microbiome and predispose women to HPV infection [330].

Regarding the association between the vaginal microbiome and cervical intraepithelial neoplasia, it has been suggested that *Lactobacillus iners* may be associated with intraepithelial neoplasia [331] and with HPV infection [332]. In HPV-positive women, this bacterium is associated with higher grades of cervical intraepithelial neoplasia [333]. *L. iners* has also been found to decrease the risk of squamous intraepithelial lesions and cervical cancer. Other studies report that various other bacteria, such as *Atopobium vaginae*, *G. vaginalis* and *Fusobacterium* spp., are also associated with cervical intraepithelial neoplasia [331,332,334,335].

#### 7.2.4. Human Microbiota and Prostate Cancer

Prostate cancer (PCa) is one of the most common cancers, and the second leading cause of death in men [262]. Its development is closely linked with the activity of androgen hormones; however, other factors, such as diet and lifestyle, may be involved [336]. A possible association between microbiota and prostate cancer has long been discussed. Chronic inflammation has been proposed to play a role in prostate carcinogenesis [337]. As *Helicobacter pylori* is known to induce chronic inflammation, inducing stomach cancer, it has been suggested that it, and other pro-inflammatory species of *Helicobacter*, may also be involved in prostate carcinogenesis [338]. Indeed, Gram-negative bacteria such as *E. coli*, *Pseudomonas aeruginosa*, and *Enterococcus* are frequently observed in urinary tract infection, and infections of *E. coli* and *Enterococcus* are characterized by higher levels of pro-inflammatory cytokines [339].

Patients undergoing transrectal prostate biopsy demonstrated increased levels of pro-inflammatory bacteria, such as *Bacteroides* and *Streptococcus*, in the rectal microbiome compared to healthy subjects. Samples from these patients are characterized by elevated numbers of bacteria associated with carbohydrate metabolism, and lower numbers of those associated with folate, biotin, and riboflavin metabolism [340]. In samples of tumoral, peritumoral, and non-tumoral tissue obtained after radical prostatectomy, the dominant phyla were Actinobacteria, Firmicutes, and Propionibacteria; however, higher levels of *Staphylococcus* spp. in tumor and peri-tumor samples [341]. Increased levels of *Propionibacterium acnes* have also been noted in samples of prostate cancer [342].

*Mycoplasma genitalium* may induce oncogenic transformation. A study of various sexually transmitted infectious agents found only *M. genitalium* to be independently associated with higher stage cancers [343,344]. In addition, elevated levels of Bacteroidetes, Alphaproteobacteria, Firmicutes, Lachnospiraceae, Propionicimonas, Sphingomonas, and Ochrobacterium bacteria have been noted in prostatic secretions of patients with PCa. Increased levels of *E. coli* were noted in seminal fluid and prostatic secretion, and lower levels in urine samples. Among patients with prostate cancer, those with benign prostatic hyperplasia were characterized by lower counts of *Eubacterium* and *Defluviicoccus* [345].

### 7.3. Human Microbiota and Lung Cancer

Lung cancer (LC) is the most common and lethal cancer worldwide, causing up to 23.1% of deaths due to all cancers [169,346]. The most prevalent type of LC is non-small cell lung cancer (NSCLC), which is subdivided into two major subtypes: adenocarcinoma (ADC) and squamous cell carcinoma (SCC). These subtypes are characterized by different biological patterns, molecular biology, and treatment strategies. Although LC is believed to be closely associated with chronic inflammation, the causes of this inflammation remain unclear [347].

The composition of the lung microbiota also remains poorly studied [348]. However, while the lung has long been regarded as a sterile space, it has recently been suggested that it does have its own microbiota, comprising approximately 2.2 × 10^3^ bacterial genomes/cm^2^ [349]. The majority of microbes that residue in the human lung belong to four phyla: Bacteroidetes, Firmicutes, Proteobacteria, and Actinobacteria, with the two main phyla in healthy lung microbiota being Bacteroidetes and Firmicutes [350]. The most abundant genera in the human lung are *Prevotella*, *Streptococcus*, *Veillonella*, *Neisseria*, *Haemophilus*, and *Fusobacterium* [351], whereas the most common in the lower respiratory system are *Pseudomonas*, *Streptococcus*, *Fusobacterium*, *Megasphaera*, and *Sphingomonas* [352].

Various diseases have the potential to cause dysbiosis of the lung microbiota, and it has been suggested that the composition of lung microbiota should be analyzed separately with regard to each [353]. It has been found that *Acidovorax*, *Klebsiella*, *Rhodoferax*, *Comamonas*, and *Polarmonas* are more frequently detected in small-cell carcinoma, but were not detected in ADC samples [354]. Another study showed a correlation between ADC and the presence of *Pseudomonas* [355], and *Capnocytophaga*, *Selenomonas*, *Veillonella*, and *Neisseria* were found in saliva samples from patients with small-cell carcinoma and those with ADC [356]. The presence of *Granulicatella adiacens*, *Enterococcus* spp., *Streptococcus intermedius*, *S. viridans*, *Streptococcus* spp., *E. coli*, and *Acinetobacter junii* was also observed in samples of lung cancer, but not in lung tissue samples from healthy controls [357].

A correlation was also found between the presence of emphysema in patients with LC and the composition of microbiota. The patients with LC and emphysema demonstrated higher levels of Firmicutes (*Streptococcus*) and Bacteroidetes (*Prevotella*) than those with emphysema only. Patients with LC cases were found to be less likely to demonstrate the phylum Proteobacteria (*Acinetobacter* and *Acidovorax*), regardless of emphysema [358].

A relationship was also found between lung microbiota composition and distant metastasis of LC in patients with SCC and ADC [353]. Among patients with ADC, those without distant metastasis demonstrated higher counts of phylum Firmicutes and genus *Streptococcus* than those with distant metastasis, suggesting that *Streptococcus* may play a protective role in LC. Among the patients with SCC, those with distant metastasis demonstrated increased levels of *Veillonella* and *Rothia* than those without metastasis. Among the patients without distant metastasis, significantly higher levels of genera *Veillonella, Megasphaera, Actinomyces,* and *Arthrobacter* were observed in the ADC patients, while among those with distant metastasis, patients with SCC were characterized by significantly higher counts of the genera *Capnocytophaga* and *Rothia* [353].

An association has been described between gut microbiota dysbiosis and lung cancer. Fecal samples from patients with LC demonstrated less diverse microbial ecosystems compared to healthy controls: lower levels of *Firmicutes* and *Actinobacteria* and a higher abundance of *Proteobacteria* and *Verrucomicrobia* compared with healthy controls [346]. In another study, significant differences in β-diversity were found between LC patients and healthy controls; the former contained lower levels of *Actinobacteria* and *Bifidobacterium* and higher levels of *Enterococcus* compared with healthy controls [359].

Many studies have noted differences in the composition of lung microbiota between LC patients and healthy subjects. The composition of lung microbiota is believed to depend also on LC subtype [360], antibiotic use [361], sample site (gut, lung, upper airways) [362], smoking status, and tumor type/grade [363,364].

In addition, the composition of lung microbiota may be associated with LC prognosis. For example, among patients with NSCLC, a greater diversity of lung commensal bacteria in unaffected distal tissues is associated with reduced disease free (DSF) and recurrence-free (RFS) survival. Higher levels of *Koribacter aceae* are associated with increased DSF and RFS, and greater occurrence of *Bacteroidaceae*, *Lechnospiraceae*, and *Ruminococcaceae*, with lower RFS and DSF [365]. Several of these results have been confirmed in animal studies [366,367].

### 7.4. Human Microbiota and Melanoma

Although melanoma is the least common cancer of the skin, it is responsible for the most deaths. The skin microbiota plays an important role in human health. Healthy human skin contains up to 10^7^ microorganisms/cm^2^, and is believed to include more than 1000 bacterial species belonging to 19 phyla [368]. The predominant genera are *Propionibacteria*, *Streptococcus*, and *Corynebacterium* [369]; however, the composition is influenced by several factors, including skin sites, sexual maturity, skin physiology, and age [370]. For example, young children demonstrate high levels of *Streptococcus*, *Rothia*, *Gemella*, *Granulicatela*, and *Haemophilus*, while adults are more associated with *Propionibacterium*, *Lactobacillus*, *Anaerococcus*, *Finegoldia*, and *Corynebacterium*. In addition, children tend to have more diverse skin microbiota than adults [371].

Few studies have examined the possible link between human microbiota and skin cancer. Nevertheless, it has been suggested that *Fusobacterium nucleatum* may be involved in the development of melanoma, and that the composition of gut microbiota may be influenced by immunotherapy in patients with melanoma [372]. Longer progression-free survival (PFS) is observed in the patients with more diverse microbial communities with higher numbers of *Faecalibacterium prausnitzii*, *Coprococcus eutactus*, *Prevotella stercorea*, *Streptococcus sanguinis*, *S. anginosus*, and *Lachnospiraceae bacterium 3 1 46FAA*, while shorter PFS was associated with higher counts of *Bacteroides ovatus*, *B. dorei*, *B. massiliensis*, *Ruminococcus gnavus*, and Blautia product. [372]. The association between the composition of human skin and gut microbiota and the development of melanoma needs further investigations.

### 7.5. Human Microbiota and Breast Cancer

Breast cancer (BC) is the most common cancer among women worldwide, and it is the second most common cause of death in women. BC is not believed to be directly associated with dysbiosis of gut microbiota.

Recently obtained results indicate that breast tissue has a specific microbiome which differs from those of other tissue. Significantly higher levels of *Prevotella*, *Lactobacillus*, *Streptococcus*, *Corynebacterium* and *Micrococcus* were detected in breast tissue samples of women free of BC who underwent either breast reduction or enhancement, compared to women with BC [373]. The composition of human breast tissue and milk microbiota are almost similar. The most abundant phyla in both breast tissue and milk are *Firmicutes*, *Actinobacteria*, and *Bacteroidetes* [374,375].

It is possible that microbial dysbiosis may play a role in the development of BC. Higher levels of *Bacillus*, *Staphylococcus*, and unclassified *Enterobacteriaceae*, *Comamonadaceae*, *Bacteroidetes* were noted in samples of breast tissue from women who underwent lumpectomies or mastectomies for either benign or cancerous tumors, compared with women without cancer [373]. In addition, breast tissue appears to have a similar microbiome between benign and invasive BC, being dominated by Bacteroidetes and Firmicutes, with an increased abundance of certain genera, including *Fusobacterium*, *Atopobium*, *Gluconacetobacter*, *Hydrogenophagea*, and *Lactobacillus* [376]. It was found also that the composition of microbiota in the urinary tract of women with BC differs from those of women without BC. However, the two groups demonstrated similar oral cavity microbiota [362,377]. Several studies have noted an association between dysbiosis and the development of BC [378,379,380,381].

## 8. Role of Gut Microbiota in Anti-Cancer Therapy

Gut microbiome dysbiosis is associated with several diseases, including cancers. Furthermore, the gut microbiota may modulate the host response to therapies administered in cancers, such as immunotherapy, chemotherapy, or radiotherapy. It is hence possible that an altered gut microbiome composition may have a beneficial effect in anti-cancer therapy [382,383,384].

Literature on anticancer therapies is very rich. There are described several therapies, such as immunotherapy, radiotherapy, and chemotherapy. There are also molecular techniques in which microRNA (miRNA), short hairpin (shRNA), small interfering (siRNA) and antisense cDNA are used. As an anticancer therapy, they are investigated (in laboratories and clinics) and used natural inhibitors, such as polyphenols, obtained from plants, and synthetic inhibitors, such as BAY876, WZB117, Glutor, fasentin, and several other (Reviewed by Pliszka and Szablewski, 2021 [385]). Unfortunately, literature on the role of gut microbiota as an anti-cancer therapy is poor. Little is known on this subject. Therefore, this paragraph, despite being very important, is very poor.

### 8.1. Immunotherapy

In 1890, William Coley injected a patient with terminal sarcoma of bacteria, referred to as “Coley’s toxin”. This procedure induced an anti-cancer effect, and saved the life of the patient [386]. Coley’s toxin contained heat-killed *Staphylococcus pyogenes* and *Serratia marcescens*. Unfortunately, this bacterial combination used in therapy was only effective in a small proportion of patients [387]. Nevertheless, Coley’s observations prompted other clinical oncologists to examine the use of microbial agents or their products as anti-cancer therapy, with varying degrees of success. In one case, the attenuated form of *Mycobacterium bovis* was administered to patients with bladder cancer [388], and *Listeria monocytes* was used as an anti-cancer therapy in pancreatic cancer [389].

Different responses to anti-programmed death-1 (anti-PD-1) therapy were observed in patients with metastatic melanoma, which were believed to be associated with the composition of gut microbiota in these patients. The responding (R) patients demonstrated higher relative abundance of *Faecalibacterium prausnitzii* compared to non-responding (NR) [390]. Patients with high counts of *Clostridiales*, *Ruminococcaceae*, or *Faecalibacterium* demonstrated increased levels of effector CD4^+^ and CD8^+^ T cells in the systemic circulation; in addition, higher levels of *Bacteroidales* in the gut microbiota caused higher levels of Tregs and myeloid-derived suppressor cells (MDSCs) [390].

The effectiveness of anti-PD-1 therapy depends also on increased levels of *Bifidobacterium longum* [391], and the responses of patients with non-small lung cancer to PD-1 therapy were found to be dependent on the composition of gut microbiota. R patients displayed a higher abundance of *Akkermansia muciniphila* in comparison with NR patients, while NR patients displayed higher levels of *Corynebacterium aurimucosum* and *Staphylococcus haemolyticus* [392]. These results suggest that, in the case of PD-1 therapy, an important role is played by the composition of the human gut microbiota [393]; however, the mechanism of this dependence remains unclear [394].

### 8.2. Chemotherapy

Gut microbiota can influence the efficacy of chemotherapy by several mechanisms, such as xenometabolism, immune reactions, and altered community structure [395], and can modify or metabolize anti-cancer drugs. It has been found that 5-fluorouracil (5-FU)-sorivudine, used in anti-cancer bi-therapy, may be metabolized by *Bacteroides* spp.; this inhibits the degradation of 5-FU, resulting in its accumulation in the blood and potentially the death of the patient. In addition, some bacteria, such as *Bacteroides* spp., *Faecalibacterium prausnitzii*, and *Clostridium* spp., can increase the toxicity of cancer therapy by producing β-glucoronidase. These bacteria can also increase the toxic effects of the topoisomerase I inhibitor irinotecan, an anti-cancer drug used in pancreatic and colorectal cancer; the drug itself is a prodrug that is metabolized into the active agent SN38. The presence of the bacteria can result in increased SN38 levels, resulting in serious diarrhea [393,396,397]. Gut microbiota have also been implicated in side effects in chemotherapy for other cancers, such as melanoma, lung cancer, colon cancer, and sarcoma.

Gut microbiota may also bestow a beneficial effect on chemotherapy. One commonly used chemotherapy drug is cyclophosphamide (CP). CP induces the translocation of commensal bacteria into secondary lymph organs, influencing the maturity of T helper 17 cells and shifting to the Th1 immune response. The immune responses facilitate a systemic anti-tumor effect [398]. The influence of gut microbiota on chemotherapy has been confirmed in a number of animal studies, particularly those based on Gram-positive bacteria, such as *Enterococcus hirae*, *Lactobacillus johnsonii*, *Ligilactobacillus murinus*, and segmented filamentous bacteria [398]. Animal studies have also revealed that the effect of platinum-based anti-cancer drugs, such as oxaliplatin or cisplatin, is enhanced by *Lactobacillus acidophilus*, restoring the anti-tumor activity of cisplatin in resistant cases [399]. Other studies have examined the interaction between microbiota and chemotherapeutic compounds [166].

### 8.3. Radiotherapy

Radiotherapy, or ionizing radiation therapy (RTX), is commonly used for patients with solid cancers. Unfortunately, RTX alters the microbiota composition.

Patients receiving RTX often demonstrate various pathologies, such as oral mucositis, enteritis, and diarrhea, which are believed to occur in response to changes in the composition of the epithelial surface microbiota. Studies suggest that irradiation-mediated intestinal toxicity is regulated by Toll-like receptor-3 (TLR3); therefore, the suppression of TLR3 signaling may decrease the gastrointestinal damage due to radiotherapy. This beneficial effect was observed after the administration of mixtures of *B. bifidum*, *L. acidophilus*, *L. casei*, *Streptococcus* spp., which are believed to reduce radiation-induced gut toxicity. Animal studies indicate that the orally administered *Streptococcus*, *Lactobacillus*, and *Bifidobacterium* spp. mixture protects animal models against toxicity due to radiotherapy [393].

### 8.4. Probiotics

Most studies are focused on the role of gut microbiota in the inhibition of cancer development, or on cancer therapy-related toxicity [400]. For example, probiotic VSL#3, which contains eight strains of lactic-producing bacteria, protects against radiation-induced diarrhea [401,402]. Several studies describe the beneficial role of probiotics in reducing side effects associated with anti-cancer therapy. The results obtained in humans have been confirmed in animal models. For example, *L. casei* may inhibit the progression of colon cancer, inducing apoptosis [403]. VSL#3 was found to induce anti-angiogenic effects and reduce inflammation, thus preventing the progression of subcutaneous hepatocellular carcinoma in an animal model [404]. However, further larger controlled clinical trials are needed on the beneficial effects of probiotics in cancer therapy [167].

### 8.5. Prebiotics

Prebiotics are typically substances intended as food for intestinal bacteria. Metabolized prebiotics act as tumor suppressive metabolites [21]. For example, polysaccharides may be metabolized to SCFAs, increasing the share of *Bifidobacterium* spp., [405], which inhibits the growth of cancer, and facilitates anti-PD-L1 efficacy. Herbal supplements may be metabolized into anti-cancer compounds. Animal studies suggest that metabolites of *American ginseng* may attenuate colon carcinogenesis; these metabolites decrease the abundance of Gram-negative bacteria, such as *Bacteroidales* and *Verrucomicrobia*, that are involved in tumorigenesis, and increase the abundance of Gram-positive bacteria, such as *Firmicutes*, which have an anti-tumorigenic role [406]. The beneficial roles of prebiotics in anti-cancer therapy have been described elsewhere [108].

### 8.6. Synbiotics

Synbiotics are the combination of probiotics and prebiotics. Few studies have been performed on the role of this mixture in cancer therapy. Patients with periampullary cancers undergoing curative or palliative treatment were treated with a mixture containing *Lactobacillus acidophilus*, *Lacticaseibacillus rhamnosus*, *Lacticaseibacillus casei*, and *Bifidobacterium bifidum* as a probiotic, and fructooligosaccharides as a prebiotic. These patients demonstrated decreased postoperative mortality and complication rates [407]. Clearly, the use of pre- and synbiotics in anti-cancer therapy requires more studies [397].

### 8.7. Fecal Microbiota Transplantation

This method, while less accepted by patients, has also been suggested for treating cancer. Animal studies suggest that FMT reduces colorectal carcinogenesis [408]; however, it is unclear whether this method reduces carcinogenesis and tumor progression in humans. Therefore, several clinical trials in which FMT is used in patients with cancer are currently ongoing [167].

## 9. Summary

Human microbiota, particularly those of the gut, influence human health and disease. These microbes play an important role in a range of processes, such as digestion, modulation of immunity, and cytokine production, and are generally beneficial to human life. On the other hand, disturbances in the balance between symbionts, commensals and pathobionts, known as dysbiosis, may influence the course of several diseases, including cancer. Such changes in the composition of gut microbiota may occur in response to several factors, such as diet, disease and pharmacology. As such, human and animal health are strongly dependent on maintaining the homeostasis of microbiota composition.

## Figures and Tables

**Table 1 ijms-22-13440-t001:** The composition and concentrations of microbiota in healthy human digestive tract with regard to body parts. The concentration of bacteria is presented as number of cells per gram of luminal content [3,14,15,19].

Region of Digestive Tract	Concentrations of Microbiota	Composition of the Microbiota Families/Genus (Species)
Mouth	10^12^	*Lactobacillus*, *Streptococcus*, *Helicobacter pylori*, *Peptostreptococcus*, *Veillonella*
Stomach	0–10^4^	*Lactobacillus*, *Streptococcus*, *Helicobacter pylori*, *Peptostreptococcus*, *Veillonella*
Duodenum	10^2^–10^3^	*Streptococcus*, *Lactobacillus*, *Bacilli*, *Actinobacteria*, *Actinomycinaeae*, *Corynobacteriaceae*
Jejunum	10^2^–10^6^	*Streptococcus*, *Lactobacillus*, *Bacilli*, *Actinobacteria*, *Actinomycinaeae*, *Corynobacteriaceae*
Proximal ileum	10^3^	*Streptococcus*, *Lactobacillus*, *Bacilli*, *Actinobacteria*, *Actinomycinaeae*, *Corynobacteriaceae*
Distal ileum	10^7^–10^8^	*Clostridium*, *Streptococcus*, *Bacteroides*, *Actinomycinaeae*, *Corynobacteria*
Colon	10^10^–10^12^	*Bacteroides*, *Clostridium* clusters IV and V, *Bifidobacterium*, *Enterobacteriaceae*, *Lachnospiraceae*, *Propionibacterium*, *Lactobacillus*, *Escherichia coli*

**Table 2 ijms-22-13440-t002:** Roles of selected human gut bacteria. For details, please see [2,10,15,20].

Bacteria	Role in Human Body
*Bifidobacterium* spp.	Produces short-chain fatty acids, improves gut mucosal barrier, decreases lipopolysaccharide levels. Some species used as probiotics.
*Bacteroides* spp.	Involved in immunity by activation of CD4^+^ T cells. Some species exclude potential pathogens from the human gut, however, others are opportunistic human pathogens.
*Lactobacillus* spp.	Produces short-chain fatty acids. Plays a role in anti-cancer and anti-inflammatory processes, produces and releases hydrogen peroxide which inhibits the growth and virulence of the fungal pathogen *Candida albicans*; some species are used as probiotics.
*Bilophila* spp.	These bacteria are involved in immunity by activation of Th1 cells. Some species are detected in perforated and gangrenous appendicitis.
*Clostridium* spp.	These species promote generation of Th17 cells, however some species of this genus are significant human pathogens, causing botulism and diarrhea.
*Roseburia* spp.	These species produce short-chain fatty acids. This genus produces butyrate, which plays several beneficial roles in human body.
*Eubacterium* spp.	These species produce short-chain fatty acids (butyrate-producing bacteria). Some species may cause bacterial vaginosis.
*Enterococcus* spp.	These species may cause urinary tract infections, bacteremia, bacterial endocarditis and diverticulitis meningitis.
*Faecalibacterium prausnitzi*	This species produces short-chain fatty acids, plays an anti-inflammatory role and boosts the immune system.
*Akkermansia muciniphila*	This species shows anti-inflammatory effects; it degrades human intestinal mucin.
*Escherichia coli*	It activates Toll-like receptors and synthesizes vitamin K_2_.
*Helicobacter pylori*	This species may cause peptic ulcer disease and gastric cancer.
*Streptococcus* spp.	Some species may cause scarlet fever, rheumatic heart disease, glomerulonephritis, pneumococcal pneumonia.
*Prevotella* spp.	The species of this genus may cause anaerobic infections of the respiratory tract and predominate in periodontal disease and abscess.
*Staphylococcus* spp.	These bacteria reside normally on the skin and mucous membranes in humans and are responsible for several common infections.
*Corynebacterium* spp.	Some species can cause diseases, such as diphtheria.
*Egerthella lenta*	This species is associated with abdominal sepsis.
*Xylanibacter* spp.	These species increases fecal short-chain fatty acid levels.
*Enterobacteriaceae*	This family includes symbionts and pathogens, such as *Salmonella*, *Yersinia pestis*, and *Shigella*.

## Data Availability

All data presented in article are available in cited articles.

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
