# Peer review of "Human Gut Microbiota in Health and Selected Cancers"

_ijms, 2021, doi:10.3390/ijms222413440_

Round 1

Reviewer 1 Report

I have no more comments. The authors revised the manuscript and it is now of high scientific quality. I recommend it to the publication in IJMS.

Author Response

Dear Reviewer

Thank you so much for taking the time, for your kind words and recommendation for our manuscript.

Reviewer 2 Report

Dear Authors,

This paper accomplished an exhaustive review of human microbiota and its relationship to various cancers.

The paper contains novel and already known data, well documented and written, plenty of discussions, and a rich bibliography. The manuscript structure is appropriately done, and I will propose only minor suggestions/corrections.

I would recommend the authors make a brief presentation of the content of their manuscript in the final part of the introduction. In addition, I think it would not be wrong if it could also refer to other studies published in 2021 on this subject.

I would also suggest adding a paragraph listing other pathologies associated with changes in the gut flora (atherosclerosis, hypertension, blood platelet function, etc.).

Author Response

Dear Reviewer

Thank you very much for your opinion, comments and suggestions.

  1. According to your suggestion, we made a brief presentation of the content of our manuscript in the final part of the Introduction. This added part is as below:

“The idea that microbes can promote the development of cancers is old, and acquired certain popularity in the late 1980s and early 1900s in the context of experiment and observations that were later disproved. Several recent studies have analyzed the oral and gut microbial dysbiosis associated with cancer risk and progression. In this review, we highlight studies analyzing role of gut microbiota in human health and diseases, especially in cancer cells. We describe the association between changes of human gut microbiota and development of selected cancers. We also discuss various strategies for microbial modulation. Additionally, we also summarize the potential role as a target in anticancer therapy”.

Also, according to your suggestion, there are added more recent studies, published in 2021, in which researchers describe the association of gut microbiota dysbiosis and cancer, as well as gut microbiota as target for anticancer therapy (References 401 – 408).

  1. In our manuscript, we discuss the association between changes in human gut microbiota and development and progression of cancers. Therefore, we think, that proposed additional diseases, are not associated with our subject. But diseases, that is proposed by Reviewer, are presented in other publications from our Chair and Department, as for example:

1) Pliszka M., Szablewski L. Human gut microbiota: Friend or Foe? OBM Hepatol. Gastroenterol., 2020; 4: 1-63

2) Szablewski L. Human gut microbiota in health and Alzheimer’s disease. J. Alzheimer’s Dis. 2018; 62: 549-560

3) Szablewski L. The role of microbiota in human health and disease. LAP LAMBERT, Academic Publishing, 2020, 342 pp.

And again thank you very much for your opinion and suggestions. We think that publication will be more friendly for readers.

This manuscript is a resubmission of an earlier submission. The following is a list of the peer review reports and author responses from that submission.

Round 1

Reviewer 1 Report

General comments

  • The authors should emphasize the novelty of this review compared to other reviews of this topic, which are very frequent when searching in the PubMed database.
  • Give the detailed aim of the review in the Introduction. The aim should indicate that this is a novelty, and indicate the differences with other reviews.
  • Authors should think of a review article written from publications of the last 5-6 years, while nearly half of the cited publications are older.
  • The Introduction and Abstract should be rewritten with reference to the topic/title.
  • All microorganisms names should be checked according to Taxonomy Browser, e.g. https://www.ncbi.nlm.nih.gov/Taxonomy/Browser/wwwtax.cgi
  • The Aims of the Journal “provides an advanced forum for molecular studies in biology and chemistry, with a strong emphasis on molecular biology and molecular medicine.” I have doubts whether the Authors deal with molecular mechanisms in the article. I leave it to the Editor's decision.

Detailed comments

  • Authors could give a short paragraph: “Methods of the review”, which databases were used, what studies from which years were taken into account, etc.
  • The title refers to cancer – what cancer? Gastrointestinal? Colon? All cancers?
  • The introduction and abstract should refer to the title, but nothing is written here about cancer. This should be rewritten.
  • Line 12: correct “hare” to “are”.
  • Lines 39-44: give some description/comments to that citations...
  • Pages 2-9 of the manuscript are unrelated to the title/topic. This should be removed or shortened and moved to the 'Introduction'
  • Are tables 1 and 2 copied from position 13 of the reference? What about autoplagiarism?

Are these tables made on the basis of authors own research or other authors studies?

Original studies should be cited.

  • Review article cannot be written on the basis of other review articles. So, all reviews in the Reference list and the manuscript should be removed (see ref. 62, 128, 154, 167, 173, 202, 235, 245, 246, 263, 298, 307, 313, 315, 340, 385, 388, 407 these are reviews) and cite original studies. If Authors have to cite any other review by way of exception, they should cite [as reviewed by ...].
  • “Paragraph 3.3. Gut microbiota and protective functions” – the subtitle is too general; maybe

Authors should refer to immunity?

  • “spp.” should be written with a normal font, not italics.
  • Authors should use new names of “Lactobacillus” throughout the whole text as it was divided into 25 new genera. Follow: http://lactotax.embl.de/wuyts/lactotax/ and https://pubmed.ncbi.nlm.nih.gov/32293557/
  • I wonder if Clostridia should be italicized? Or maybe it should be written as clostridia ...?

Lines 287, 351 it should be “Clostridium”.

  • Authors should use the correct term “microbiota” or “microbiome” instead of inaccurate “microflora” related to human body as it has nothing in common with plants (flora).
  • Paragraph “8. Role of gut microbiota in anti-cancer therapy”- this paragraph was described very poorly, while the authors wrote about it in the aim of the article ...
  • Summary is also poorly written. Authors should give paragraph: Conclusions and future perspectives as this topic is very developmental, interesting and promising.
  • Line 1666 it should be “nucleatum” not “nucelatu”. Follow: https://www.ncbi.nlm.nih.gov/Taxonomy/Browser/wwwtax.cgi

Author Response

Dear Reviewer

Thank you very much for your review of manuscript, comments and suggestions. According to your suggestion, there are introduced several changes, as below.

General comments

  • The authors should emphasize the novelty of this review compared to other reviews of this topic, which are very frequent when searching in the PubMed database.
  • We think, that our review article emphasizes the novelty in comparison with other reviews. In our article there are presented different results, sometimes contrary and controversial, obtained by several authors. Therefore, our article may be as a source of information, older or recent, for readers. 
  • Give the detailed aim of the review in the Introduction. The aim should indicate that this is a novelty, and indicate the differences with other reviews.
  • The aim of the review is presented in the Abstract. Because title of article is changed (it was added “Health”), altered is also aim. Recent version of aim is: “The aim of this review is to describe the role gut microbiota in health and the association……”
  • Authors should think of a review article written from publications of the last 5-6 years, while nearly half of the cited publications are older.
  • There are 242 articles of the last 5-6 years. This is >58% of all cited publications. In many journals, also belonging to MDPI, it should be minimum 50% of recent publications. Therefore, our list of References is correct.
  • The Introduction and Abstract should be rewritten with reference to the topic/title.
  • Because in title of our article is added word “Health”, we think that The Introduction and Abstract is adequate and don’t need changes.
  • All microorganisms names should be checked according to Taxonomy Browser, e.g. https://www.ncbi.nlm.nih.gov/Taxonomy/Browser/wwwtax.cgi
  • Because our article is review article, we must use names of microorganisms that are used in cited publications.
  • The Aims of the Journal “provides an advanced forum for molecular studies in biology and chemistry, with a strong emphasis on molecular biology and molecular medicine.” I have doubts whether the Authors deal with molecular mechanisms in the article. I leave it to the Editor's decision.
  • You are right, that title of Journal is International Journal of Molecular Sciences, but we sent our manuscript to Special Issue: Gut dysbiosis. Molecular mechanisms and Therapies. Therefore, there are, for example, information on influence of several metabolites of microorganisms on human processes. We describe also, how these metabolites may change course of metabolism, or genetic information, causing cancer disease. We think, that this is molecular biology/medicine.

Detailed comments

  • Authors could give a short paragraph: “Methods of the review”, which databases were used, what studies from which years were taken into account, etc.
  • We think, that this information isn’t necessary. But in list of cited articles (References) above mentioned information is included. Any databases were used, because in cited publications, if it was necessary, authors gave these data, as well as in cited articles are included years of publication.
  • The title refers to cancer – what cancer? Gastrointestinal? Colon? All cancers?
  • We describe 13 main types of cancers and many subtypes. Should we to mention all cancers in title?
  • The introduction and abstract should refer to the title, but nothing is written here about cancer. This should be rewritten.
  • It was discussed in point 4 (General comments).
  • Line 12: correct “hare” to “are”.
  • This error is corrected.
  • Lines 39-44: give some description/comments to that citations...
  • We don’t know, how should we comment these sentences? These sentences don’t need description or comments.
  • Pages 2-9 of the manuscript are unrelated to the title/topic. This should be removed or shortened and moved to the 'Introduction'
  • Because title is changed, these pages are necessary and are in correct place.
  • Are tables 1 and 2 copied from position 13 of the reference? What about autoplagiarism?
  • Position 13 of the reference may be named autocitation, not autoplagiarism. There are many differences between Tables 1 and 2 presented in reviewed manuscript and in presented in book (position 13). Please compare these tables. In book (position 13) are cited names of all, 26 authors. Because these tables differ, they contain also different information, therefore, this isn’t autoplagiarism. It is also presented source of information (position 13). This position was cited in the case of mentioned tables, because in book, these tables contain more information.

Are these tables made on the basis of authors own research or other authors studies?

Original studies should be cited.

  • Review article cannot be written on the basis of other review articles. So, all reviews in the Reference list and the manuscript should be removed (see ref. 62, 128, 154, 167, 173, 202, 235, 245, 246, 263, 298, 307, 313, 315, 340, 385, 388, 407 these are reviews) and cite original studies. If Authors have to cite any other review by way of exception, they should cite [as reviewed by ...].
  • These 18 reviews contain very important information, based on original publications. In these reviews authors wrote also suggestions, conclusions and so on. Therefore, they are cited. We think, that isn’t necessary to write “reviewed by…” In few cases title of cited article contains word “review”
  • “Paragraph 3.3. Gut microbiota and protective functions” – the subtitle is too general; maybe
  • There are several other information that should be described in details, such as protective function of microbiota. But, manuscript is so long, and few points are shortened and described only in general. It was suggested, that microbiota play different roles, not only as pathogens, but also protective function.

Authors should refer to immunity?      

  • “spp.” should be written with a normal font, not italics.
  • This was corrected.
  • Authors should use new names of “Lactobacillus” throughout the whole text as it was divided into 25 new genera. Follow: http://lactotax.embl.de/wuyts/lactotax/ and https://pubmed.ncbi.nlm.nih.gov/32293557/
  • As we wrote above, because manuscript is review article, based on other publications, we must use names that are in original publication.
  • I wonder if Clostridia should be italicized? Or maybe it should be written as clostridia ...?
  • It was changed as Clostridium.

Lines 287, 351 it should be “Clostridium”.

           This error was corrected.

  • Authors should use the correct term “microbiota” or “microbiome” instead of inaccurate “microflora” related to human body as it has nothing in common with plants (flora).
  • Term “microflora” was used in in the cases: Intestinal Microflora Transplantation (mode of therapy), titles of articles and so on. If so, term microflora was used in the case of bacteria, not human body.
  • Paragraph “8. Role of gut microbiota in anti-cancer therapy”- this paragraph was described very poorly, while the authors wrote about it in the aim of the article ...
  • Because literature on this subject is poorly.
  • Summary is also poorly written. Authors should give paragraph: Conclusions and future perspectives as this topic is very developmental, interesting and promising.
  • However, this point is very interesting, but, as we wrote above, manuscript was too long.
  • Line 1666 it should be “nucleatum” not “nucelatu”. Follow: https://www.ncbi.nlm.nih.gov/Taxonomy/Browser/wwwtax.cgi
  • This error is corrected.

Reviewer 2 Report

This review needs to be restructured. The title state a revision between the gut microbiota and cancer, but the reader has to read 10 pages of non-related information until finding the first introduction on this topic. Apart from this, I have found many inaccuracies and misleading information. For example in section 3.2, lines 122-123, authors state "The gut microbiota is known to synthesize primary bile acids (BAs), cholic acid (CA) 122
and chenodeoxycholic acid (CDCA) from cholesterol derived from the human liver [29].". This is not true. The gut microbiota does not synthesise primary bile acids. They are synthesised by the liver and then converted into secondary bile acids by the gut microbiota.

Author Response

Dear Reviewer

Thank you very much for your review, opinion and suggestion.

According to your suggestion were done changes as below:

This review needs to be restructured. The title state a revision between the gut microbiota and cancer, but the reader has to read 10 pages of non-related information until finding the first introduction on this topic. Apart from this, I have found many inaccuracies and misleading information. For example in section 3.2, lines 122-123, authors state "The gut microbiota is known to synthesize primary bile acids (BAs), cholic acid (CA) 122
and chenodeoxycholic acid (CDCA) from cholesterol derived from the human liver [29].". This is not true. The gut microbiota does not synthesise primary bile acids. They are synthesised by the liver and then converted into secondary bile acids by the gut microbiota.

* Title of article was changed on “HUMAN GUT MICROBIOTA IN HEALTH AND CANCER”, therefore above mentioned 10 pages are necessary for readers.

* You are right, that this sentence is wrong. Primary bile acids are produced by liver, whereas microbiota produce secondary bile acids. This sentence was corrected and now is “The gut microbiota is known to synthesize secondary bile acids (BAs), deoxycholic acid (DCA), ursedeooxycholic acid (UDCA) and litocholic acid (LCA) from cholesterol derived from human liver” Grüner and Mattner, Int. J.Mol. Sci. 2021, 22, 1397. Thank you very much for information on this error.

* This manuscript was sent earlier to Native Speaker, and proofreading was done. We found only few errors (changed letter, lack of space between words). Therefore, we think that now manuscript don’t need additional proofreading.

Round 2

Reviewer 1 Report

  • The authors should emphasize the novelty of this review compared to other reviews of this topic, which are very frequent when searching in the PubMed database.

We think, that our review article emphasizes the novelty in comparison with other reviews. In our article there are presented different results, sometimes contrary and controversial, obtained by several authors. Therefore, our article may be as a source of information, older or recent, for readers.

The above answer is the opinion of the Authors. The novelty of this review should be reasonably scientifically emphasized by the Authors in the manuscript.

  • All microorganisms names should be checked according to Taxonomy Browser, e.g. https://www.ncbi.nlm.nih.gov/Taxonomy/Browser/wwwtax.cgi   

Because our article is review article, we must use names of microorganisms that are used in cited publications.

I understand that you should duplicate the errors of other Authors? This is not a scientific approach ...

  • The Aims of the Journal “… provides an advanced forum for molecular studies in biology and chemistry, with a strong emphasis on molecular biology and molecular medicine.” I have doubts whether the Authors deal with molecular mechanisms in the article. I leave it to the Editor's decision.

You are right, that title of Journal is International Journal of Molecular Sciences, but we sent our manuscript to Special Issue: Gut dysbiosis. Molecular mechanisms and Therapies. Therefore, there are, for example, information on influence of several metabolites of microorganisms on human processes. We describe also, how these metabolites may change course of metabolism, or genetic information, causing cancer disease. We think, that this is molecular biology/medicine.

I still maintain my opinion that the article is not written at the level of molecular knowledge.

  • The title refers to cancer – what cancer? Gastrointestinal? Colon? All cancers?

We describe 13 main types of cancers and many subtypes. Should we to mention all cancers in title?

I am sorry to say that this answer seems impolite... The authors did not understand my question...

  • Lines 39-44: give some description/comments to that citations…

We don’t know, how should we comment these sentences? These sentences don’t need description or comments.

The point is that these sentences are detached from the rest of the text, they should be introduced somehow ... I read the introduction, and suddenly some quotes appear ... What should I do with them?

  • Are tables 1 and 2 copied from position 13 of the reference? What about autoplagiarism? Are these tables made on the basis of authors own research or other authors studies? Original studies should be cited.

Position 13 of the reference may be named autocitation, not autoplagiarism. There are many differences between Tables 1 and 2 presented in reviewed manuscript and in presented in book (position 13). Please compare these tables. In book (position 13) are cited names of all, 26 authors. Because these tables differ, they contain also different information, therefore, this isn’t autoplagiarism. It is also presented source of information (position 13). This position was cited in the case of mentioned tables, because in book, these tables contain more information.

I am sorry that the Authors think that I am accusing them of being auto-plagiarism. I only asked a question, because I have no access to those sources and cannot check ...

  • Review article cannot be written on the basis of other review articles. So, all reviews in the Reference list and the manuscript should be removed (see ref. 62, 128, 154, 167, 173, 202, 235, 245, 246, 263, 298, 307, 313, 315, 340, 385, 388, 407 these are reviews) and cite original studies. If Authors have to cite any other review by way of exception, they should cite [as reviewed by ...].

These 18 reviews contain very important information, based on original publications. In these reviews authors wrote also suggestions, conclusions and so on. Therefore, they are cited. We think, that isn’t necessary to write “reviewed by…” In few cases title of cited article contains word “review”

All the references I have mentioned include a ‘review’ in the title…Writing a review from other reviews is unacceptable. This is how I was taught by my professors. In all the review articles that I have published, reviewers have always pointed out this. Writing a review from other reviews confirm the lack of an idea, original scientific thought and the lack of novelty, as I mentioned earlier ...

  • Authors should use new names of “Lactobacillus” throughout the whole text as it was divided into 25 new genera. Follow: http://lactotax.embl.de/wuyts/lactotax/ and https://pubmed.ncbi.nlm.nih.gov/32293557/

As we wrote above, because manuscript is review article, based on other publications, we must use names that are in original publication.

The conclusion is that if we apply this practice indefinitely, we will never start using a new nomenclature ...

  • Paragraph “8. Role of gut microbiota in anti-cancer therapy”- this paragraph was described very poorly, while the authors wrote about it in the aim of the article ...

Because literature on this subject is poorly.

The reviewer is well aware of this, but maybe it is worth mentioning it in the manuscript ...?